# Field Work's Optimization for the Digital Capture of Large University Campuses, Combining Various Techniques of Massive Point Capture

**José Javier Pérez, María Senderos, Amaia Casado and Iñigo Leon ***

Department of Architecture, University of the Basque Country UPV/EHU, Plaza Oñati 2,
20018 Donostia-San Sebastián, Spain; josejavier.perez@ehu.eus (J.J.P.); maria.senderos@ehu.eus (M.S.);
amaia.casado@ehu.eus (A.C.)
**\*** Correspondence: inigo.leon@ehu.eus; Tel.: +34-943-01-7192

**Abstract:** The aim of the study is to obtain fast digitalization of large urban settings. The data of two university campuses in two cities in northern Spain was captured. Challenges were imposed by the lockdown situation caused by the COVID-19 pandemic, which limited mobility and affected the field work for data readings. The idea was to significantly reduce time spent in the field, using a number of resources, and increasing efficiency as economically as possible. The research design is based on the Design Science Research (DSR) concept as a methodological approach to design the solutions generated by means of 3D models. The digitalization of the campuses is based on the analysis, evolution and optimization of LiDAR ALS points clouds captured by government bodies, which are open access and free. Additional TLS capture techniques were used to complement the clouds, with the study of support of UAV-assisted automated photogrammetric techniques. The results show that with points clouds overlapped with 360 images, produced with a combination of resources and techniques, it was possible to reduce the on-site working time by more than two thirds.

**Keywords:** LIDAR; TLS; UAV; point cloud; 3D modelling

## 1. Introduction

The global environmental situation is critical, with the depletion of natural resources, global warming and $CO_2$ emissions leading to greater environmental awareness [1]. The building sector alone accounts for 18.4% of total anthropogenic greenhouse gas emissions [2]. It is necessary for cities and the built environment to fulfill their potential to enhance energy efficiency [3]. Digitalization, defined as the development and deployment of digital technologies and processes, is considered crucial for the required transformation of the construction industry to improve productivity according to the report of World Economic Forum [4].

In this sense, our research took two main lines from an architectural perspective: On the one hand, the optimization of the digital capture of a constructed setting [5], and the use of digital 3D models for environmental assessment of urban settings [6]. In early March 2020, we commenced a research project that linked both areas of research. We were then faced with the crisis caused by the COVID-19 pandemic, which led to the state of national confinement decreed on 14 March 2020 in Spain [7], as was the case in many other countries. The first task of the project consisted of the digital capture of two large university campuses. This process usually entails a great deal of on-site work. Given the situation of confinement imposed by COVID-19, it was very difficult to spend long periods of time on site to take data readings. Mass capture of points in the urban settings was necessary, to obtain multiple data (coordinates, distances, surface areas, angles, temperatures of facades, etc.), and this had to be performed at breakneck speed with the fewest possible resources.

Several options had to be studied to find a combination of resources and techniques. The decision was made to start the work by using Light Amplification by Stimulated Emission of Radiation (LiDAR) points clouds, captured by manned Airborne Laser Scanning (ALS), carried out by the government in both cities (a resource that is free in many countries). The LiDAR ALS points clouds are an easily accessible and cheap resource, but their accuracy and performance need to be complemented by other capture techniques. The research conducted with the LiDAR ALS clouds, access to which is instant, open and free, including analysis, editing and optimization, enabled the additional techniques needed to complete the final points cloud of each campus to be estimated. In this, particular, case study, Terrestrial Laser Scanning (TLS) was considered to be the fastest option to complement the final points cloud of the urban area, extending the study to the support of automated photogrammetric techniques assisted by Unmanned Aerial Vehicle (UAV).

Design Science Research (DSR) is a methodology that can provide solutions for research through the use of three-dimensional models. It employs techniques such as case-studies, data collection and document analysis, among others, and centers on creating and optimizing artifacts to improve processes and their operative performance [8]. With this methodology, research objectives are approached more pragmatically than in explanatory scientific investigation [9].

After the digital capture of the urban setting in 3D, the second phase of the project focused on the environmental assessment of the campuses. The tool used was Neighborhood Evaluation for Sustainable Territories (NEST), a tool based on life cycle evaluation methodology (ACV) [10,11]. Although some results of this assessment have already been published [12], this article does not focus on this phase, it only shows the minimum information necessary to give context to the research as a whole.

This article focuses on the results of the optimization work performed on the digital capture of the two university campuses up to when the 3D simulation models are obtained. The conclusions include the result that after previously working with the LiDAR ALS points cloud of the Government of Navarra, the normal on-site reading period with TLS, estimated at 52 days for the campus of Pamplona, could be reduced to 7 days. The combination of devices, software and applications that were used made it possible to reduce the scanning time with overlapped 360 image capturing by more than 75%, in comparison to customary scanning times on the market. We found that even with such a fast capture time we were able to obtain errors of 1 mm, with a strength and overlap that was accepted as valid by the processing software. Therefore, this article could be of great benefit to the scientific community engaged in work of this nature, since it would help them to be more efficient and make effective use of resources.

The article is structured as follows: The introduction consists of two sections. Section 1 contextualizes the research. Section 2 describes the case study, including the current state of research into different capture techniques. Section 3 describes the Methods and Materials. Section 4 describes the results of generation of LiDAR point clouds, modelling and simulation. Section 5 presents the Discussion, and the article ends with the final conclusions.

## 2. Digital Survey of Large Urban Areas in a Short Time, Study Cases

The case studies focus on two universities in northern Spain: the campus of the University of the Basque Country (UPV-/EHU) in Donostia-San Sebastián (DSS) and the University of Navarra (UNAV) in Pamplona (Figure 1).

The UNAV campus has an area of approximately 113 ha, which includes large open grassy areas, slightly wooded areas and a riverbed flanked by a dense mass of trees. The buildings cover only 6.8% of the total area. The UPV/EHU university campus in DSS has a much smaller area (approximately 18 ha) and a much higher building density, with a much lower proportion of green areas. As previously mentioned, the aim of this work is, to carry out the field work in the shortest time possible, with the fewest number of resources.

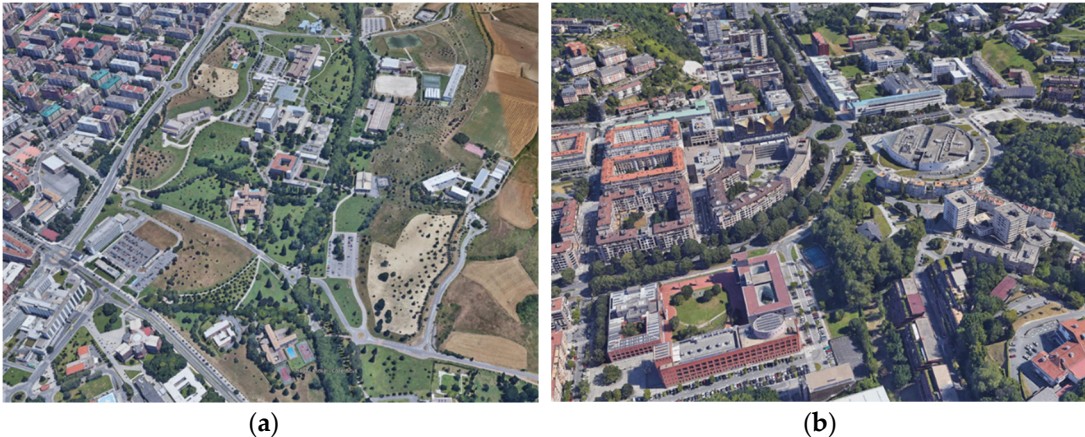

(**a**)                                                        (**b**)

**Figure 1.** (**a**) Aerial view of the UNAV university campus in Pamplona; (**b**) aerial view of the UPV/EHU university campus in DSS.

The modeling phase in NEST requires a prior graphic survey of the current state [13]. Depending on the reason for using the model, it will require a sufficient degree of precision to accurately determine the geometric configuration of buildings and their surroundings [14]. Although the NEST 3D model does not require excessive precision [15], the survey does require definition of the building envelopes [16], marking the number of floors, window configurations and opaque elements, and of the spaces occupied by roads and highways; green spaces and trees must also be defined [17], so that forest biomass can be calculated [18–20].

Given the limited time available for carrying out fieldwork, the use of massive point capture techniques will allow highly precise geometry to be obtained in digital format in a very short time. The combined use of digital geometric data collection techniques [21], is currently the most effective procedure for conducting a precise geo-referenced architectural survey [22]. In such cases, it includes a topographic survey with a total station [23], terrestrial laser scanner and short-range photogrammetry assisted by an RPA (Remotely-Piloted Aircraft) or UAV [24–27].

LiDAR technology also makes it possible to acquire massive amounts of 3D geospatial information in urban scenarios [28–31]. LiDAR technology measures the properties of reflected laser pulses to determine the range of a distant object [32]. That range is obtained by measuring the delay time between transmission of a laser pulse and detection of the reflected signal [33]. Due to LiDAR's ability to generate 3D data with high precision and spatial resolution, a new era is opening for the development of research objectives such as the one presented [34]. LiDAR scanning can be classified in four categories: Satellite-based Laser Scanning (SLS), Airborne Laser Scanning (ALS) [35–38], Mobile Laser Scanning (MLS) and Terrestrial Laser Scanning (TLS) [39]. ALS is ideal for large areas of cities [40,41]; it can be conducted by UAVs or by manned aircraft, which usually fly at higher altitudes and capture larger areas than UAVs, though the latter are cheaper and less polluting, among other advantages [42–44]. SLS data points can be tens of meters apart and the respective point clouds are therefore unsuitable for extracting geometries from urban features such as buildings or masses of trees [45]. TLS data has the highest point density and can be used to specify data for those urban elements at individual level [46–48]. Some publications claim that TLS sometimes has poor mobility and occlusion issues that make it difficult to collect data on an urban scale. When TLS is not effective, MLS has been used in some research, such as for collection and analysis of information on trees in urban areas [49,50].

Various techniques have been studied, and the method that best fit the objectives of this work involved the combination of different resources: ALS LiDAR clouds captured by public administrations, point clouds captured using TLS and, finally, point clouds produced from automated photogrammetry assisted by UAVs.

## 3. Methods and Materials

In this section the technologies enabling the massive point capture to engender the point cloud of each university campus in a very short time [5], are explained, specifying the techniques, methods and materials used. The research design is based on DSR. DSR is used to design and assess manmade artifacts meant to resolve real-world problems [51]. This method helps find practical solutions for common problems affecting design, with a view to achieving expected results [8], and employs computer-based tools to streamline processes [51]. When a problem is associated to a physical object, the respective solution may appear as a 3D model, plan or drawing; when it requires optimizing an action, the solution may take the form of new digital software or be developed as a flowchart diagram [51]. DSR includes other non-habitual forms for conveying knowledge, such as models or constructs [52]. That is why DSR expresses knowledge based on different formats not commonly found in other scientific investigations, such as, for example, 3D models, architectures, design theories or principles and artifacts [53]. Two main activities are put forward in design science: to build the solution and to evaluate it [54]. The construct is a stage within the process of creating an artifact that can be used to resolve a specific problem. The evaluation is the action that must validate how effectively that artifact serves the purpose for which it was created. This is precisely what is going to be conducted in this investigation: to make, evaluate and optimize a 3D digital model of the urban area in the form of a point cloud with 360 image that contains all the information needed to achieve the project's objectives. The construct stage must necessarily be iterative and incremental, as the evaluation phase will endow it with the feedback needed to optimize the solution. DSR enables relevant problems to be resolved based on applied research appearing in some scientific investigations linked to architecture [55].

### 3.1. Analysis of LiDAR Clouds Obtained by Public Services

In Spain, different public services offer that LiDAR data, thereby simplifying the data capture process for these kinds of projects, with the respective point clouds obtained using manned aircraft. The great advantage of these clouds is that they are public and can be consulted for free. In the case of point clouds obtained by ALS systems, the latest sensor technology has significantly increased the number of laser light beams per square meter. As a result, the density of the point clouds generated during the data collection process shows a range of between 12–30 points/m$^2$, compared to the range of 1 point/m$^2$ obtained by previous sensors. The Chartered Community of Navarre was one of the first European regions to apply LiDAR technology using these new sensors, specifically the Leica Single Photon LiDAR (SPL100). A sensor is able to capture light particles with a laser light beam that can be divided into a $10 \times 10$ matrix, operating in practice as 100 sensors in parallel, each of which is captured by an independent channel of the detector. The experimental flights were conducted in 2017; after processing and classifying the data obtained using Artificial Intelligence (AI) techniques, it was possible to cover an area of 10,391 km$^2$, generating meshes of $1 \times 1$ km, with a point cloud density of 14 points/m$^2$ and a precision of 20 cm on the XY axis and 15 cm on the Z axis.

The Navarre government's partial LiDAR clouds from 2017 were initially used, and a point cloud of the entire UNAV Campus in Pamplona was composed. Different cuts to the cloud were performed at strategic points; the precision and suitability of the cloud were also analyzed to study the combination of techniques (Figure 2).

The campus cloud was segmented to form detailed sets of buildings and check their geometry with the density value of 14 points/m$^2$. The definition of that façade would apparently suffice to obtain a 3D simulation model (Figure 3).

Dimensional checks of the result will, subsequently, be carried out to ascertain which buildings need to complete the point cloud with other LiDAR techniques.

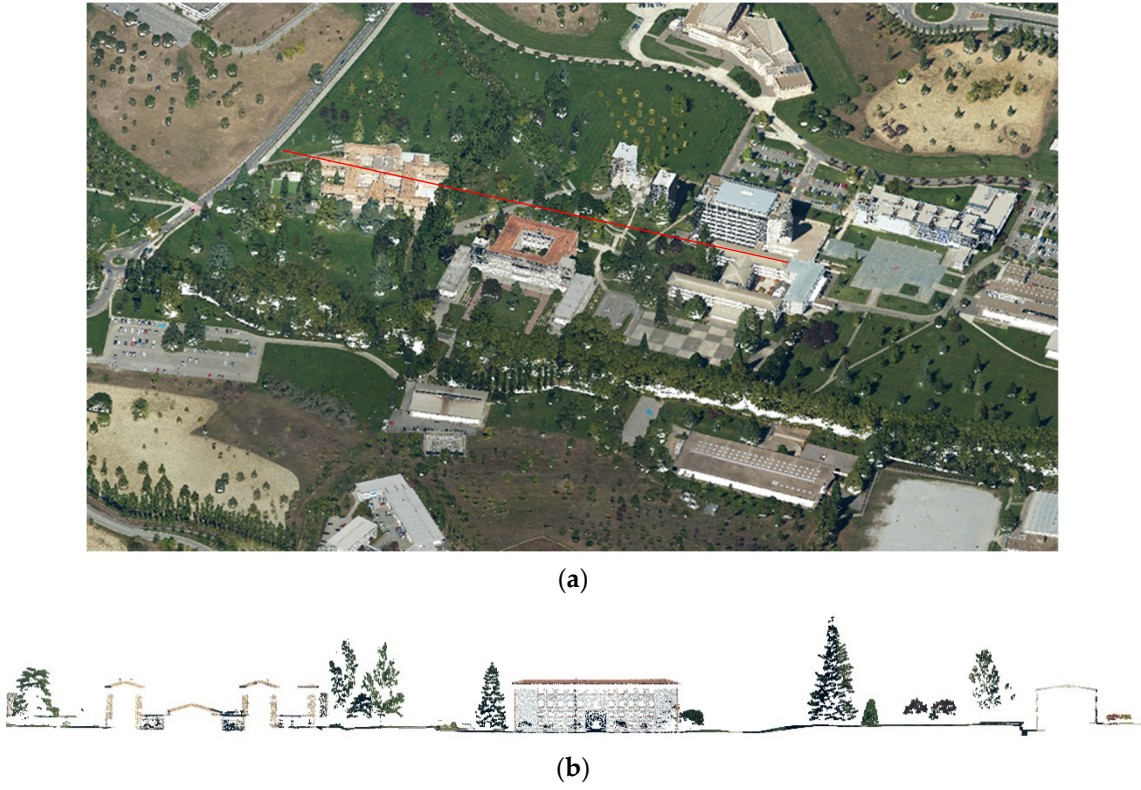

(**a**)

(**b**)

**Figure 2.** LiDAR 2017 point cloud, density: 14 points/m$^2$. UNAV campus: (**a**) 3D color cloud; (**b**) vertical section of the cloud through the north façade of the central campus building.

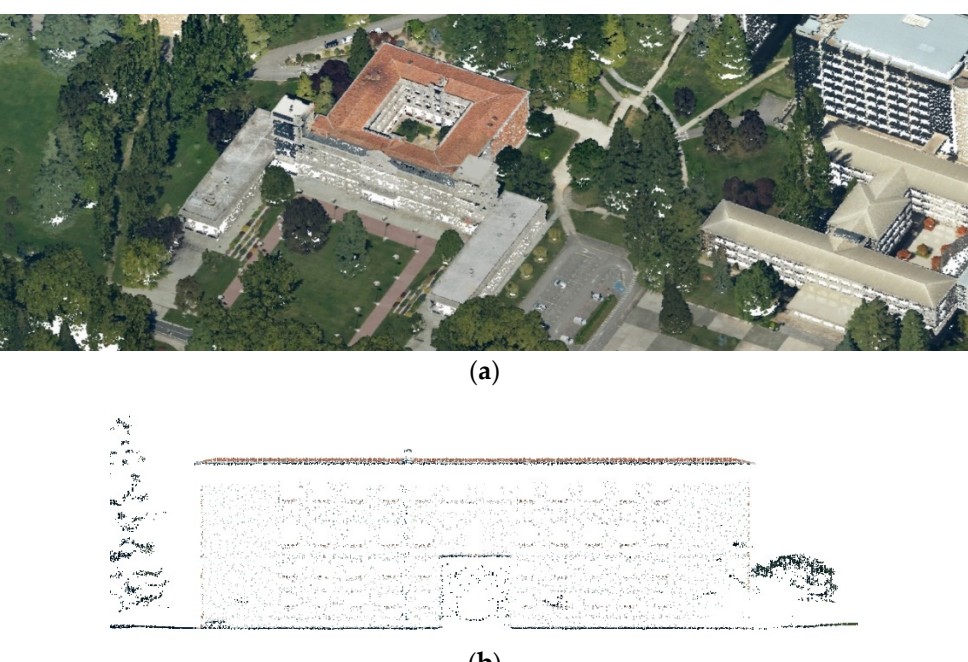

(**a**)

(**b**)

**Figure 3.** 2017 LiDAR cloud density: 14 points/m$^2$. UNAV campus: (**a**) 3D detail of the central campus building; (**b**) north façade of the same building.

To achieve the objectives of this research, other data of interest included the estimated approximate volume of the campuses' forest biomass; detailed tests of tree masses were accordingly conducted. It was thereby possible to verify another fundamental characteristic of these new sensors, used in 2017. They enable capture of the terrestrial relief, devoid

of any artificial and/or natural element other than the ground (DTM—Digital Terrain Model), and of the earth's surface with all built or natural bodies on it (DSM—Digital Surface Model). The use of specific wavelengths enables penetration between tree masses, capturing the lower ground level, which facilitates the height measurement of those masses. In this regard, an example of the capture of plant masses at the UNAV university campus in Pamplona is shown below. Cloud cuts were conducted in wooded areas. In the case of the densely populated vegetation zone, the scanner's ability to penetrate the tree mass and record the ground level is observed [14,42], (Figure 4). Although the trees' compactness makes it difficult to fully record the respective mass, the information captured allows for approximate measurements of the height and volume of the vegetation, with a precision that can be estimated to the nearest decimeter.

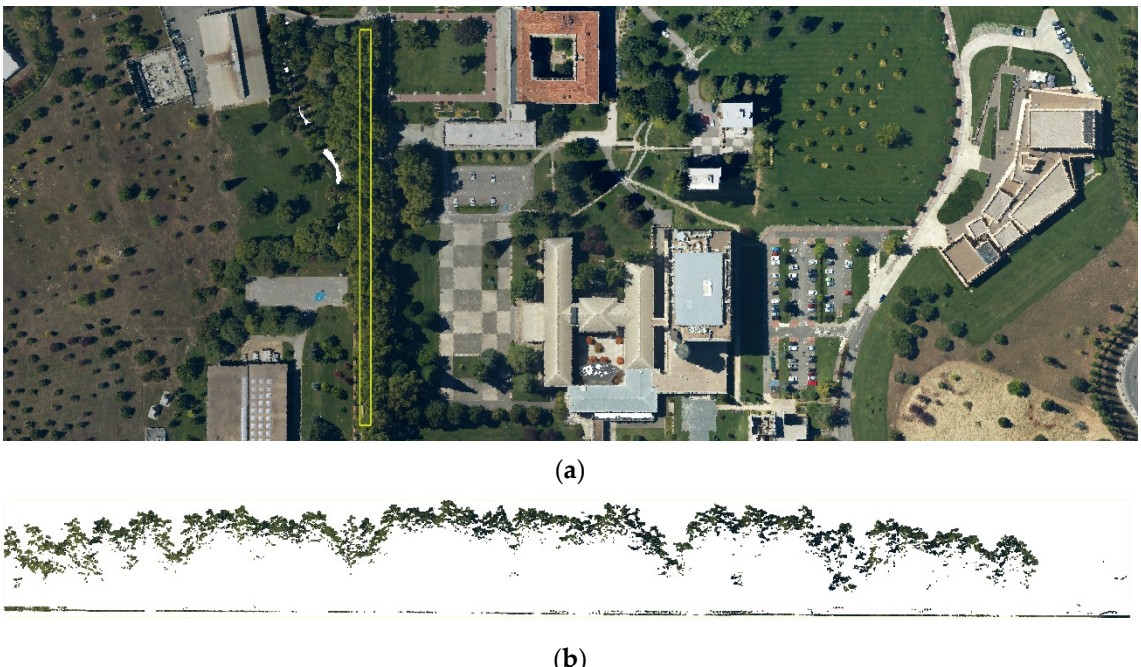

(**a**)

(**b**)

**Figure 4.** UNAV campus, vegetation strip example: (**a**) 2017 LiDAR Cloud Plan, density: 14 points/m$^2$; (**b**) profile of tree mass and level of ground under that mass.

After analyzing the possibilities of the LiDAR cloud of the UNAV campus in Pamplona, with a density of 14 p/m$^2$, the LiDAR clouds currently available for the territory of Gipuzkoa province, where the UPV/EHU campus is located in DSS, are analyzed. In that province, LiDAR clouds captured in 2012 and 2017 are currently available (Figure 5).

The 2012 LiDAR flight presents meshes of 2 × 2 km with a density of 1 point/m$^2$, while the 2017 LiDAR flight presents 500 × 500 m meshes with a density of 2.2 points/m$^2$. As in the UNAV's case, a partial cloud was created for the entire UPV/EHU campus, for both the 2012 and 2017 clouds. Partial sections of the campus buildings and trees were likewise made to compare the accuracy and usefulness of the clouds.

Although specific measurements of these LiDAR clouds will be presented in the Section 4 with the analysis of these two examples from the UPV/EHU campus, several limitations can be appreciated. In the 2012 cloud, total height of buildings could be obtained; however, the volumes of buildings are not intuited, nor are vertical stripes marked. In addition, the profile of tree masses presents excessively isolated points, and it cannot be determined whether they are masses or specific trees. In the 2017 cloud of the same campus, building heights are correctly appreciated, volumes are marked with vertical stripes and there is greater definition of tree masses. Even with this definition of 2.2 points/m$^2$, building façades could not be modeled nor could biomass volumes be calculated, unlike what was seen in the cloud of the UNAV campus.

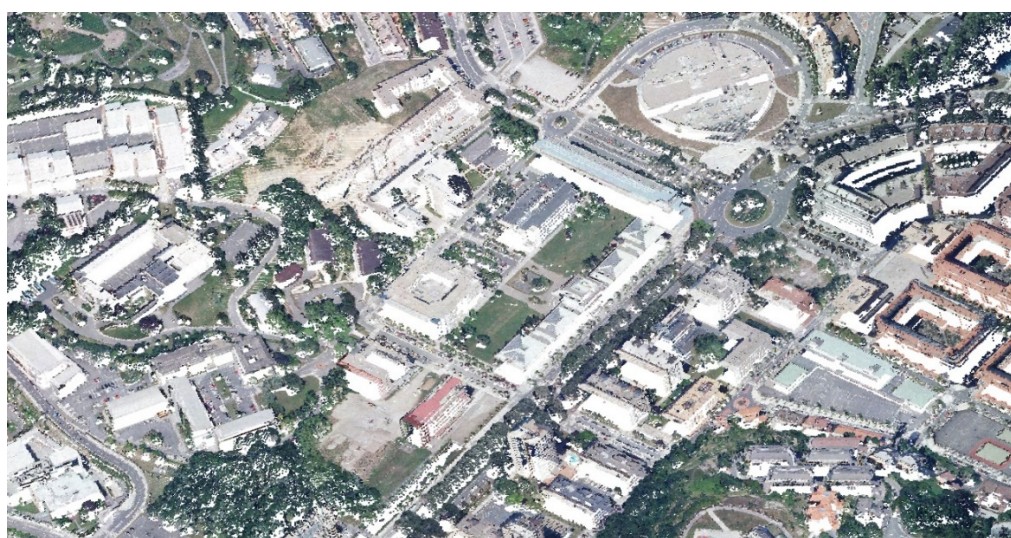

**Figure 5.** 2012 flight LiDAR point cloud of the UPV/EHU university campus in DSS. Cloud density: 1 point/m$^2$.

The precision of these clouds will condition subsequent data collection at the two campuses to complete the point cloud that allows the 3D simulation model to be achieved.

### 3.2. Data Collection to Complete the LiDAR, MLS and TLS Clouds

TLS and MLS technologies make it possible to obtain highly accurate point clouds. However, managing those technologies to capture urban environments of a certain size requires in-depth study to ensure that they can be effective, sustainable and relatively cheap. MLS scanners are generally much more expensive than TLS scanners. However, depending on the urban environment, they can reduce execution times and therefore be a more efficient option. In any case, the scanning of such areas must be planned very well so that the point clouds are not excessively dense and can be handled by standard hardware.

Regarding the MLS LiDAR options, some work has opted for models, such as the Leica Pegasus, which allows reality to be captured from a vehicle, train or ship. It is an expensive option that requires very specific capture conditions for large areas in cities, though it is very useful for capturing linear infrastructures. Was not considered due to the characteristics of the two campuses. One option that was tested is the Leica BLK2GO handheld scanner, which captures moving images and point clouds in real time, using SLAM (Simultaneous Localization and Mapping) technology to record their course through space [56,57]. That scanner combines dual-axis LiDAR, a 4.3 Mpx 360° panoramic viewing system, a 12 Mpx high-resolution camera for detailed photos and an inertial measurement unit that enables self-navigation, capturing 420,000 pts/s with a capture range of 0–25 m. System performance based on SLAM technology offers 6–15 mm relative accuracy and 20 mm absolute positioning accuracy for maximum range [5]. Considering the size and characteristics of the campuses, the scanning process with this device was ruled out both due to execution times and the excessive amount of information that would be captured.

Bearing in mind the accuracy of the LiDAR clouds discussed in the previous point, TLS was deemed the fastest, most efficient and least problematic option to complete the clouds, although it is true that specific locations can be complemented with captures made using UAVs. The geometric data capture technique using terrestrial laser scanning allows this capture to be performed quickly and expeditiously, capturing a large amount of information at very high speed, from medium and long distances and with a high degree of accuracy. The generated high-density point cloud can be supplemented by 360° panoramic photography taken at each scan position. The point cloud with the overlapping 360 image, makes it possible to configure a three-dimensional visual environment wherein it is feasible to make millimetric measurements and develop virtual visits. Some scanners also have

a built-in thermographic camera that can discern the temperature of each of the millions of points captured when scanning. The temperature of the facades is of special interest in this type of project in which energy improvement is proposed by means of passive solutions such as energy-minded reform of building façades. The five methodological stages followed when scanning the two campuses with TLS will be described next.

### 3.2.1. Survey of Control Points in UTM Coordinates Using a Total Station

The main objective of the control point system is to obtain a three-dimensional digital model of the geo-referenced survey in absolute UTM coordinates. Obtaining a geo-referenced model is not an essential requirement in cases where the data collection procedure is accomplished by laser scanning, since a local coordinate system can be used. However, this information's implementation in the photogrammetric processing ensures greater accuracy of the three-dimensional digital model. Moreover, the control points guarantee the rigor and accuracy of the data processing and facilitates the union between different captures. The materialization of those points is performed using adhesive targets for fixation on the different supports, in the form of rigid plates of variable size. The control points, located on the ground, are permanently referenced by topographic nails for their maintenance during execution of the work.

The checkpoint system layout follows the following criteria:

- Link checkpoints, for joining point clouds corresponding to the different laser scans. Located on vertical walls of the façade and meant to cover the maximum possible width, both vertically and horizontally. Some of these control points, if located on vertical roof faces, can help facilitate the union between the data captured by laser scanner and the UAV-assisted photographic capture;
- Checkpoints on the roof, the topographical targets that we usually place to give more precision to the photogrammetric work of the UAV in relation to the work of the laser scanner. We placed them at the ends of the roofs at different heights (especially on the campus buildings that had flat roofs with several volumes of different heights). That way some targets are captured by the 3D laser scanner and by the UAV, and this facilitates the union between points clouds;
- Checkpoints on the ground, common to both data collection procedures for later integration. Situated in such a way that they are recorded by both laser scanning and UAV-assisted photographic capture.

### 3.2.2. Scanning Plan

For the scanning process to be effective, it is recommended that a prior study be conducted of the scan positions for the set to be captured. With respect to university campuses, the capture focused on two important aspects:

- The exterior survey of each campus's buildings, both in cloud format and in a 360° image, to obtain a multitude of data so that the 3D simulation model could be created from the office without having to venture into the field;
- Registration of the buildings' external environment, where the main aim focused on capturing the green areas with more or less trees, to record and measure the amount of available forest Biomass.

Because the situation generated by the COVID-19 health crisis meant that movement was very limited, optimizing the fieldwork was vitally important, it meant to reduce the scanning process to the shortest time possible. To be efficient, it was essential to study the campuses' cartography before proposing a scanning plan. The UNAV campus in Pamplona, has terrain with large slopes in some areas and that this needs to be taken into account when preparing the scan position plan. If a plan is drawn up without taking the slopes into account, as if the terrain were flat, the distances between scan positions are horizontal projections of the actual distance. The research focus in this article is on reducing the number of scan points as much as possible, therefore, the need to take the slopes into account in the scan plan is an important one.

As mentioned above, the two campuses have very different characteristics that affect the scanning plan. The Donostia campus is a relatively flat urban campus without large areas of trees; it is therefore practically possible to arrange its scanning plan using an orthophoto. The work was divided into three different areas that cover the entire UPV/EHU campus. However, the Pamplona campus is overwhelmingly complex. The extent of the campus and the large number of buildings are a challenge that cannot be practically covered in a short time by a terrestrial laser scanner. In many cases, the terrain's unevenness exceeds the height of buildings, which are located at very different heights and very far apart. Furthermore, the medium-height vegetation and above all the large trees in many cases prevent the capture of many building façades' geometry. If it were not for the high quality of the 2017 LiDAR clouds, this would be an overwhelming task in a short period of time, even using a scanner as versatile as the RTC 360. On the UNAV campus, simplified CAD planimetry was used to conduct a prior study of possible scan positions. Considering the high number of scan positions and the large area covered by the cloud take, it was decided to divide the work into seven zones. However, this depends a great deal on the power of the computer that will be used when processing the clouds.

### 3.2.3. Laser Scanning, Pre-Processed in the Field with Mobile Devices

The building façades on the different campuses were scanned, as well as the exterior environments, with special attention given to green areas and vegetation. In the survey of the buildings the main objective was to measure the façades' dimensions, differentiating the sizes of window openings and opaque surfaces. A minimum of three scans were conducted for each façade of the campus buildings. Diagonal scans were also performed to capture the internal faces of the façades. Depending on the distance between buildings, the remaining scans were distributed on the ground. A color point cloud treatment was conducted, since a 360° panoramic view was also captured at each scan point. Two terrestrial laser scanners were used: an RTC 360 and a BLK 360, both from Leica Geosystems (Figure 6). They stand out due to three characteristics: they are extremely light, it is not necessary to spend time levelling and they capture 360° spherical images in HDR in a short time, generating a color point cloud [23]. We have worked with scanners of other brands, and we are aware that in such cases capture with a 360 HDR image under 8 min is complicated. The two Leica scanners used in this study enabled the project objectives to be achieved. The characteristics and features of the devices are very important in enabling us to obtain the results mentioned in the article.

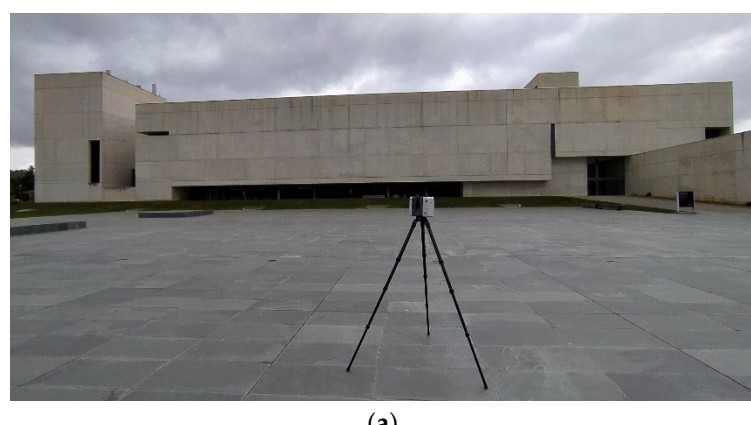 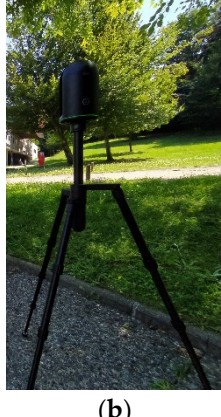

(**a**)       (**b**)

**Figure 6.** Scanners used: (**a**) RTC 360; (**b**) BLK 360.

The first one was the BLK360, which has a registry range of 60 m and gathers 360,000 points/second. It has a pair of special features that make it very interesting. One is that it has a 360 thermal camera that is very useful for sustainability and energy efficiency issues. The other is that it is very small and only weighs 1 kg. It acted as a complement in

some specific tasks for the second scanner, which was the main device used in field capture. The second device, the RTC 360, has a registry range of up to 130 m, which enables it to cover large areas. It is especially interesting from an energy perspective, because it has VIS technology that enables it to automatically register device displacement without targets from one scanning point to another, so that the partial points clouds are registered in a special location related to the other scans. The device's measuring rate is up to 2 million points/second, and for high resolution, (3 mm@10 m), it can scan in 1:42 min without HDR. Bearing in mind that medium resolution is often enough in many cases, a point cloud with 360 image can be obtained in a scan of less than 2 min. We can safely say that this is "little time" in comparison to other scanners. In fact, we have checked and found that the 360-image captured with other 3D laser scanners in 8 min is of poorer quality than the one captured in one minute with these scanners.

Unlike other scanners, the BLK and the RTC enable pre-processing work to be performed on site, via a mobile device (tablet or mobile phone), thanks to the Leica Cyclone FIELD 360 application. While the scanner is capturing points, we can check the results obtained and move forward with the processing work, uniting the points clouds of each scan position. These scanners transmit a Wi-Fi network that enables the mobile device to be linked to the scanner, so that all the scanner data can be transferred in real time to the Cyclone FIELD 360 application. Any tablet or mobile phone that uses iOS or Android can be used for this purpose. Represents an advance that further streamlines work, besides enabling the campus digitalization work to be evaluated, optimized and validated to obtain the points cloud in 3D. It also enables the basics of the DSR method to be complied with: implement, evaluate and optimize.

The scan positions pre-processing app has several work tabs. In the "map" format the clouds are joined in a plan following the "cloud-to-cloud" method, and the accuracy of the union in plan, section and perspective can be consulted. The union must always be conducted between two nearby point clouds, which will be shown in two different colors to facilitate the process (usually orange and blue-cyan). The "360" section allows immersion in each scan point, to view details. Lastly, a specific cloud or the assembled set of clouds can be viewed in 3D (Figure 7).

### 3.2.4. Information Processing—Point Clouds, 360° Images

The processing software used, Leica Cyclone Register, presents an environment similar to that of pre-processing but allows working with more tools to make the final result more accurate. Any operation carried out with the pre-processing software can be reversed; it enables visualization of the error, overlap and strength of each union between clouds. The process is simplified into four steps associated with four tabs: Import Data, Review and Optimize (the pre-processed data), Finish (and generate different files) and Produce an Accuracy Report. After analyzing the joints' suitability, they must be reviewed and optimized according to the link error, overlap and force parameters. As already mentioned, measurements of distances, surfaces and angles can be made in the process. The cloud of the complex or that of a specific scan position can be displayed in 3D. To optimize the ensemble, this process is conducted "cloud by cloud" by analyzing two nearby clouds (Figure 8). At the end of the process, the error is displayed as a graph. The process can be repeated as many times as necessary.

Once the result has been achieved with the expected accuracy, the process ends. The software allows the results to be exported to different file formats, and, also, generates a log report with a large amount of process data to validate the work carried out.

### 3.2.5. Results and Reports

After completing the connection between clouds and verifying that the quality of the connections is appropriate for the project objectives, the software enables the generation of an exhaustive precision report.

The cloud can also be exported to multiple formats (LGS, e57, RCP, etc.) so that the project partners can work in their environments and with their software. These files are usually exported to intermediate software such as Recap, which allows the cloud to be imported into common modeling programs (Revit, Sketchup, etc.). The partial clouds of the different zones were exported to LGS format to view and extract the data from the free Jetstream Viewer. The whole set was not joined, to avoid working with an excessively dense file.

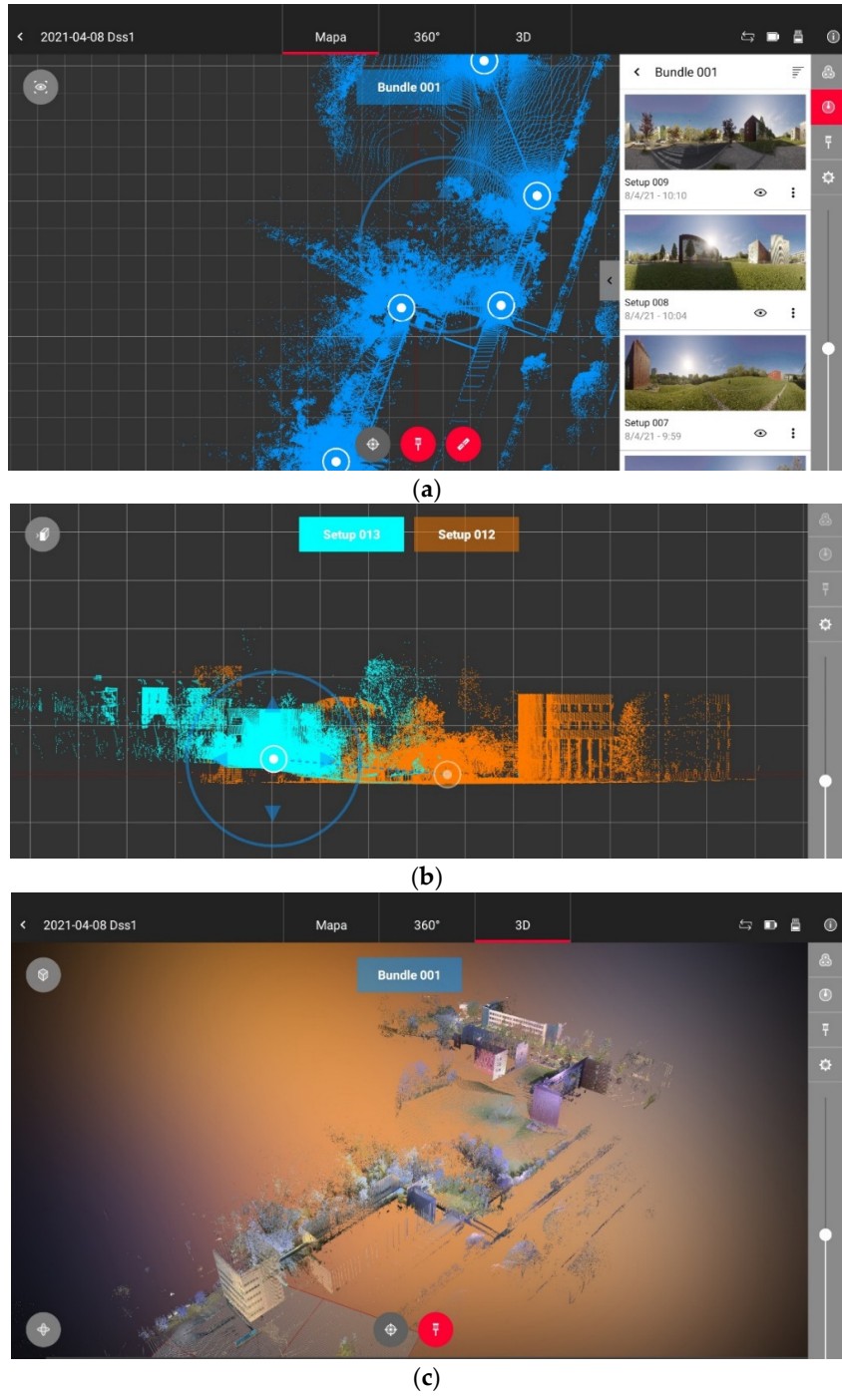

**Figure 7.** Cloud pre-processing with the Leica Cylone Field 360. DSS Campus, Zone 3: (**a**) union of the clouds in plan and 360° panoramic images; (**b**) inion in "cloud-to-cloud" section between two clouds; (**c**) 3D cloud join preview.

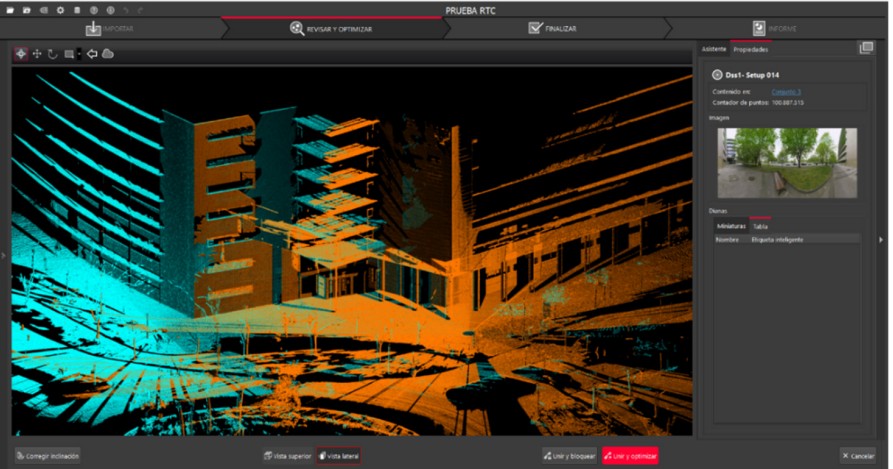

**Figure 8.** Editing and optimization of point clouds "cloud-to-cloud", using Cyclone software. DSS Campus, Zone 2.

### 3.2.6. Visualization—Obtaining Data to Feed the Model

The display of the model has two graphic options: one to display the point cloud (in color if 360° images were obtained) and the other to display the model in 360° panoramic image format. In the second option the point cloud is overlaid in the background with all the data at each point. So, if we want to check point coordinates, distances between two points, surfaces in m², angles or surface temperatures at a point, this data is extracted quickly and easily. Furthermore, the interface allows navigation in the model with different options (flight, orbit, etc.), so that the campus can be "visited" virtually with options not possible in the real model. With this resource, part of the field work can be transferred to the office, from where the campus can be visited and perfectly measured in a very short time, allowing to measure even places that would be inaccessible in the real model. The results can also be viewed using other 3D viewing software such as the Leica TruView viewer or other external software such as Autodesk ReCap.

### 3.3. Capture by UAV and Structure from Motion (SfM) Photogrammetric Processing

The use of UAV-assisted photogrammetry makes it possible to complete the digital model of the survey [58], in cases where relevant geometric data exists outside the laser scanner's range [59]. Examples include roofs [60], eaves and, generally, all horizontal surfaces above the origin of the projected laser light beam [18]. The data on building roofs is largely meant to obtain the surface of the buildings' current solar panels, which is relevant information for the simulation of the 3D model with NEST.

Two options can be incorporated in a UAV: an MLS device that can instantly obtain LiDAR clouds or a camera that captures data through automated photogrammetry. The first option requires higher performance of the device. An interesting option is the BLK2FLY, a UAV with built-in MLS that is very light. It features a GrandSLAM sensor fusion of LiDAR, radar, cameras and GNSS for full scan coverage, plus optimized flight paths. It also has advanced obstacle avoidance for greater flight safety. Although it is a very interesting option, because it can be integrated in the same flow and software proposed in this article, it was ruled out for this work because it is more expensive. A similar high-priced asset is the DJI Zenmuse L1, with LiDAR built into the UAV. The advertising for the Zenmuse L1 mentions a coverage of 400 Hectares in one sole flight at 100 m height and a speed of 13 m/s. Some unofficial practical trials, mention a lower capacity for capturing 80 or 90 Ha. in one single flight. Other sources state that the combined capture between LIDAR and the photographic series, at 300 feet and with a GDS of 10 cm/pixel, would cover little more than 5.5 Ha. in a 27 min flight. Such data is doubtlessly very promising, but we should not forget that the recent appearance in the market of these new tools means that

for now at least there are insufficient practical trials and contrasted scientific studies on their performance. We will have to wait a while for further studies and contrasted research to see the results in the appropriate publications.

To supplement the clouds previously obtained, the occasional support of automated photogrammetry by means of UAV was chosen. UAV-assisted photogrammetry or data collection using RPAs and SfM photogrammetric processing [27], facilitate data collection, due to the process's simplicity and the breadth of the working range [61]. The constant innovation process of this new technology has developed intelligent flight planning and control applications that enable semi-automatic design and execution of data collection. These applications very considerably reduce the time devoted to fieldwork for data gathering, even for work ranges of large areas [43,44]. The flight altitude determines the width of the data capture range [62], and, at the same time, its resolution or the GSD value (ground sample distance or effective pixel size of the model's bitmap), in such a way that the respective value is proportional to the working range and inversely proportional to the captured data's resolution. If the capture does not require high levels of precision, that is why a balanced relationship between capture range and GSD value can facilitate an efficient data collection process, providing optimal results. The processing of the photographic take enables a textured geometric mesh to be obtained, which can be exported to a point cloud format, and thus integrated in the point cloud previously generated in the processes described above (ALS and TLS) [63].

The biggest drawback when using this technology concerns the flight restrictions imposed by the respective air safety agencies and the administrative authorizations required to carry out the flights. If you are an official drone operator in Spain with the appropriate permits, you can fly over almost any area, but processing such permits can take a very long time, and this can be a problem when you are planning for data readings. On many occasions the work and delays involved in this administrative procedure ended up affecting the choice of one type of technology over another. At the UPV/EHU campus it was therefore not considered necessary to complement the cloud with this technology, while on the Pamplona campus it was used to complete specific elements in areas with few valid references for TLS.

An automated flight configuration simulation for data collection from a 1 ha area is shown below. To improve the geometric characterization of the vertical planes, the combined use of one nadir shot (gimbal tilt—90°) and two oblique shots (gimbal tilt—45°) was configured (Figure 9 and Table 1).

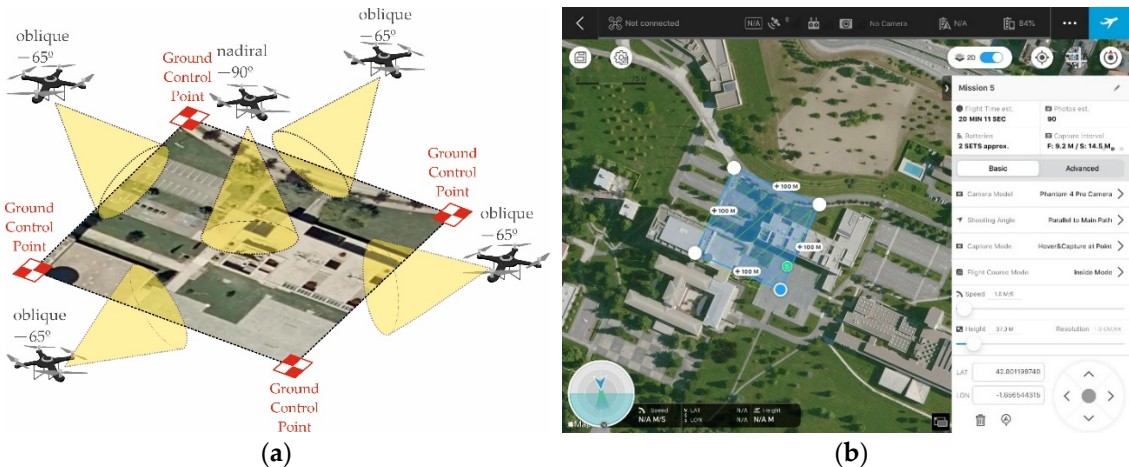

**(a)**          **(b)**

**Figure 9.** Automatic flight configuration for a surface of 1 ha in the UNAV university campus in Pamplona: (**a**) UAV shots and Gimbal inclinations; (**b**) data in the DJI-GS Pro app.

**Table 1.** Automated flight configuration chart for the coverage of a 1 ha area and with photogrammetric model resolution of 1 cm/pixel.

| Shot Coverage | Front Overlap between Photos | Side Overlap | GSD Resolution | Photographs per Shot | Flight Duration per Shot | Nadiral Shot −90° | Oblique Shot −65° | Total Flight Time |
|---|---|---|---|---|---|---|---|---|
| 1 ha | 75% | 75% | 1 cm/px | 90 | 20 min. | 1 | 4 | 100 min. |

### 3.4. Modeling and NEST-Sketchup

The point cloud can then be imported into 3D modeling software to generate the model (Autodesk Revit, Sketchup, etc.). The process involved two phases. Initially, the residential buildings in the campus were modeled (Figure 10); finally, the university buildings were modeled in more detail.

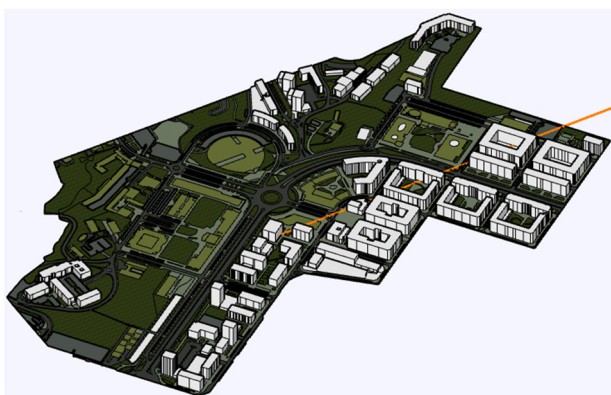

**Figure 10.** 3D model for the environmental assessment. UPV/EHU DSS campus, phase 1.

From the final cloud obtained, accompanied by the merged panoramic image, the data and measurements needed to complete the model were obtained. With these measurements the model was generated in NEST-Sketchup and the different reform scenarios were simulated.

With the NEST tool an environmental assessment can be performed from the LCA perspective using different elements such as buildings, means of transport or urban lighting. An urban plan for a totally new urban area can be analyzed, or an evaluation of an urban renewal area. This tool was developed based on a doctoral thesis [64]. It is a plug-in for the SketchUp 3D modeling software widely used in architecture and urbanism. The interface is therefore very intuitive, based on a graphic environment that allows theoretical aspects to be related to the built reality. NEST directly obtains the evaluation from the 3D model of the urban area studied and calculates it based on a set of indicators created from a scientific perspective.

The 3D model with information has to be created to carry out the evaluation. To do so, the first thing that needs to be conducted is to model the geometry of the buildings and of the roads, parking, green spaces, etc.:

- Model from the plan, take measurements from the 360 image with the cloud superimposed, create the buildings first and then the rest of the urban layout;
- Model from the points cloud inserted in Sketchup, which is a faster process.

Then the model needs to be fed with the information necessary to run the environmental evaluation. NEST considers four main elements in the urban scope: building features, ground typologies (green space, road, etc.) and public amenities such as urban lighting and user transport. Most tools that evaluate such environments do not assess all the LCA stages set out in standard ISO 14040 [65]. Some consider only the operative energy use phase or the product phase. NEST assesses the environmental impact of more stages, as indicated in (Table 2).

**Table 2.** Building life cycle stages defined by NEST.

| | Product Phase (A1–3) | Transport (A4) | On Site Processes (A5) | Maintenance (B2) | Replacement (B4) | Operational Energy Use (B6) | Operational Water Use (B7) | End-of-Life Phase (C1–4) |
|---|---|---|---|---|---|---|---|---|
| Buildings | - | - | - | - | - | - | - | - |
| Land use | - | - | - | - | - | - | - | - |
| Infrastructure | - | | | | - | - | - | |
| Mobility | - | - | N/A | N/A | N/A | - | | - |

### 3.5. Materials for Method

The usual resources that were used and/or tested in the research developed for this project will be summarily listed. Resources that could have been used to obtain the ALS LiDAR clouds will not be included, since they were captured and processed by different public services, and it is not precisely known which resources were used.

### 3.5.1. Material Resources Required

To apply the techniques described, material resources or devices that allow this data to be captured in the field must be used. The characteristics of such devices have been described in previous publications [5,23], so it is not necessary to include them (Table 3).

**Table 3.** Material resources used in the massive point capture.

| Equipment | Trademark | Model | Use |
|---|---|---|---|
| Total Station | Leica Geosystems | TCR-407 | Establishment of Control Points in UTM coordinates. |
| Terrestrial Laser Scanner (TLS) | Leica Geosystems | RTC-360 | Massive capture of points by laser scanning. |
| | Leica Geosystems | BLK 360 | Massive capture of points by laser scanning. Thermographic capture. |
| UAV-RPAs | DJI | Phantom 4 pro | Massive capture of points through photographic series. |
| Tablet | Samsung | TAB S6 LITE 64 Gb RAM | RPAs flight and laser scanning operation interface. |

### 3.5.2. Software and Hardware

To complete the entire work process, several types of software are necessary. The software used for development of the phases described in the work process for LiDAR clouds will be summarily specified: referencing of the work, capture and pre-processing of different field scans, processing of the different clouds obtained in each scan position to obtain the final set, visualization of the final product and accurate measurement of the work conducted. Some software is specific to the scanner brand used in the fieldwork, while other software can work with files generated by various brands (Table 4).

**Table 4.** Software used in the massive point capture.

| Work Phase | Software | Scan Data Import | Import Formats | Export Formats |
|---|---|---|---|---|
| UTM referencing | Survey Office [66] | - | ASCII, LandXML | ASCII, LandXML, GIS/CAD |
| Scanning and Pre-processing | Field 360 * [67] | - | Direct Wifi connection with the 3D scanner or UAV automatic import to tablet | Direct export to Leica Cyclone REGISTER 360 via Wi-Fi network. |
| | DJI-GS Pro (UAV) [68] | | | Photographs in the SD of the UAV |
| Processing | Cyclone REGISTER 360 * [69] | Only Leica | LGS, PTX, PTS, E57, etc. | Almost any format: cloud of points and geometries. Accuracy reports. 360 images. |
| | PointCab [70] | Independent | XYZ, E57 | Point information, CAD formats Autocad (DWG), E57 PTS RCP/RCS |
| | ReCap Autodesk [71] | Independent | XYZ, E57 | |

**Table 4.** *Cont.*

| Work Phase | Software | Scan Data Import | Import Formats | Export Formats |
|---|---|---|---|---|
| Results visualization | TruView * [72] | Only Leica | LGS | Almost any format: cloud of points and geometries. Accuracy reports. 360 images. |
| | Jetstream Viewer * [73] | Only Leica | LGS | Point information, CAD formats (free) |
| | ReCap Autodesk [71] | Independent | XYZ, E57 | Autocad (DWG), E57 PTS RCP/RCS |

\* Leica.

## 4. Results

Although part of the results were shown in the previous section to enable better and more graphic understanding of the methodological process, this section focuses on the partial results that will allow the 3D simulation model to be obtained. Some quantitative data of the LiDAR cloud, the error from the union of the TLS clouds and the UAV flight operation will be shown. In addition, graphic results of the cloud will be explained, so that the way data is extracted to generate the 3D model can be checked. Finally, some values from the environmental assessment will be presented, though these are not the direct objective of this publication.

### 4.1. ALS LiDAR Clouds

In this section, the results of the ALS LiDAR point cloud produced using the LiDAR of public services involved in the project will be analyzed. Two main aspects will be studied: the cloud's suitability for modeling the façades of campus buildings, considering that the buildings' volumes, the windows and opaque parts of façades must be defined; and the suitability for calculating biomass volumes by making sections of trees masses in the cloud.

As for the campus buildings, twenty buildings were evaluated at the UPV/EHU campus in DSS, while 31 were evaluated at the UNAV campus in Pamplona [12]. Before determining the scanning plan strategy for TLS, the 51 buildings had to be analyzed by making partial sections of the ALS cloud. Below is an example of measurements made in a building on the UNAV campus, where the cloud has a density of 14 pts/m$^2$ (Figure 11).

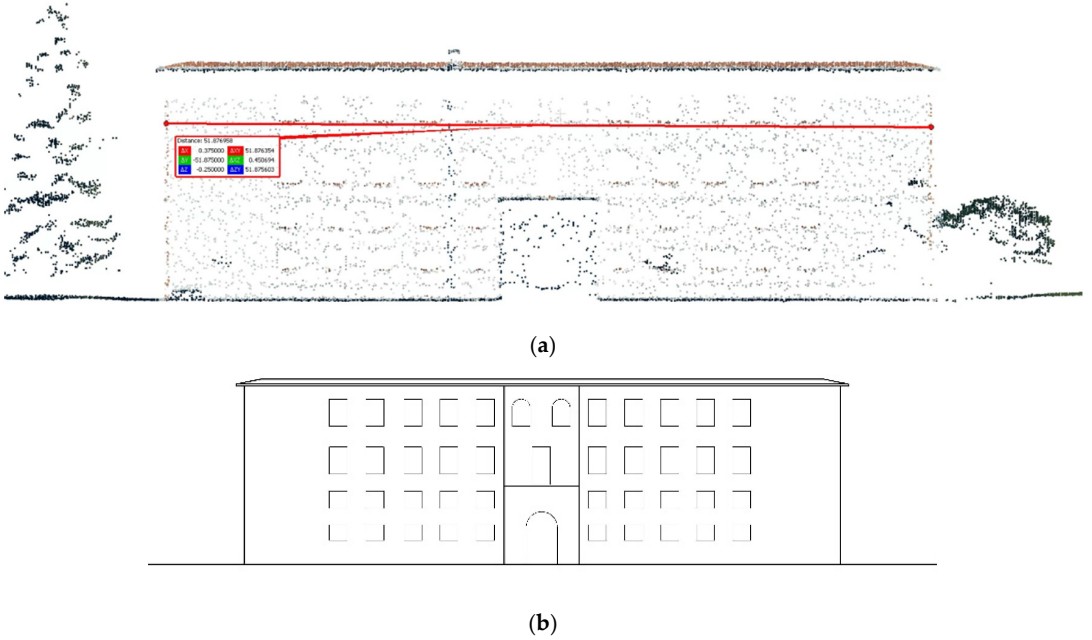

(**a**)

(**b**)

**Figure 11.** 2017 LiDAR cloud, density: 14 points/m$^2$. UNAV campus: (**a**) measurements on the point cloud of the campus's central building; (**b**) graphic survey of the same building façade.



As an example, a series of basic measurements were made to calculate building height, façade surface and openings on the north façade of the central building at the UNAV university campus in Pamplona. The point density value of 14 p/m$^2$ allows approximate measurements of part of the building elements to be obtained, whose accuracy may sometimes be sufficient for the 3D simulation model. The results obtained in the example are shown below (Table 5).

**Table 5.** Example of façade measurement table in the ALS LiDAR cloud.

| Facade Length | Height to Eave | Facade Surface | Window Dimensions F0/F1 | Window Dimensions F2/F3 |
|---|---|---|---|---|
| 51.90 m. | 15.50 m. | 804.45 m$^2$ | 1.55 × 1.35 m. | 1.55 × 2.30 m. |

It was thus possible to obtain measurements for all buildings on the UNAV campus; if any additional measurement was necessary, supplementary measurements from the cloud obtained with TLS techniques was used. On the UPV/EHU campus, the LiDAR clouds in Gipuzkoa province were not able to obtain the same results. It was only possible to obtain measurements of façades (height and width), but not of window opening sizes. To conduct that, a more exhaustive capture had to be performed with TLS. The data used to model buildings on the UPV/EHU campus in DSS was extracted directly from the LiDAR clouds obtained with TLS techniques.

The results were also analyzed to obtain the campuses' approximate forest biomass volume. The measurement of an isolated tree or vegetation element will be used here as an example, starting with analysis of the suitability of the LiDAR clouds at the UPV/EHU campus. In Figure 6, the 2012 cloud barely shows points of vegetation or soil. The 2017 cloud of the two campuses is analyzed in comparison (Figures 12 and 13).

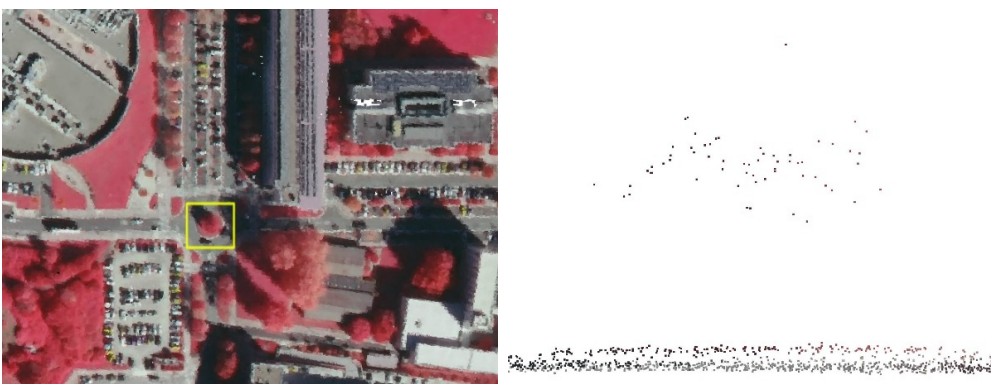

**Figure 12.** Isolated vegetation element. 2017 LiDAR cloud, density: 2.2 points/m$^2$. UPV/EHU campus in DSS.

In the cloud in Figure 12, the terrain's configuration can be observed, though it is difficult to measure tree volume. After making several measurements in that LiDAR cloud, the estimate obtained has a maximum precision of 30 cm in the XY axes and 20 cm in the Z axis. In contrast, it was verified that in the LiDAR cloud of the UNAV campus in Pamplona, the tree masses have enough precision to make measurements. In Figure 13 the tree's height is exactly 22.45 m.

Many publications present multiple ways to calculate tree volumes [74]. Considering the requirements of the NEST evaluation software, calculations have been performed in two ways: for isolated trees such as the one in the example, their volume is assimilated to a cone, cylinder or sphere [75]. In Figure 13b it is assimilated to a cone. The tree's total height and the base of the branches are measured; the volume of the cone is then calculated. For continuous masses, such as in Figure 4, partial sections of the cloud are used, and the approximate volume is extracted considering the contours of the mass. To

calculate the final volume of the campus's biomass, that tree data must be completed with the volumes of green areas associated to the terrain's green surfaces. In the case of the UNAV campus, its total area was calculated as being 1,547,278 m$^2$, with a green space surface area of 1,082,210 m$^2$, accounting for 70% of the total. At the UPV/EHU campus in DSS, the total area of the campus was estimated to be 565,140 m$^2$, with a green space area of 168,816 m$^2$, accounting for approximately 30% of the total. Numerous publications have explained different ways of calculating forest biomass [75–77]. With this data it was possible to feed the NEST model to carry out the evaluation.

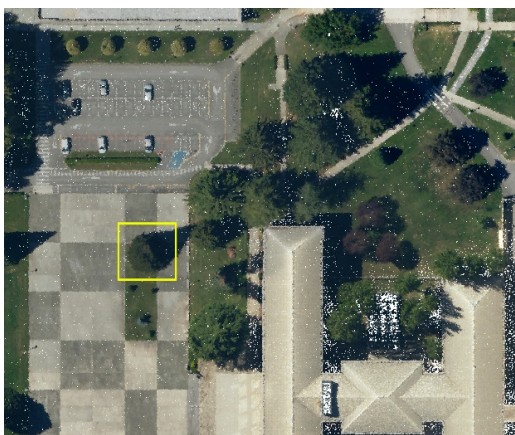 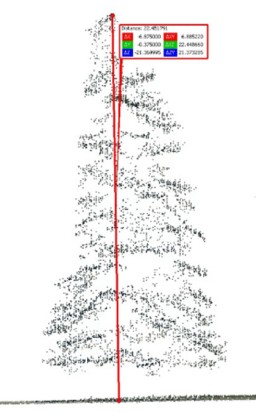

**Figure 13.** 2017 flight LiDAR point cloud, density: 14 points/m$^2$. Isolated vegetation element where the complete section of its mass is observed, as well as level of the ground.

### 4.2. TLS LiDAR Clouds

The measurements that could not be obtained from the ALS LiDAR cloud were made from the cloud supplemented by TLS techniques. All kinds of geometric data (lengths, angles, areas, etc.) and thermal data of the points in the cloud can be extracted from that cloud using laser scanning. At the end of this section, in the visualization part of the resulting clouds, some examples of quantitative results in cloud measurements can be seen.

Although we have shown that the field work in 7 days enabled enough results to be obtained to enter all the necessary data in NEST, we shall show how the previous estimate of 7 days was calculated. The basis for everything is to draw up different scanning plans, placing the scan positions at different distances. The lesser the distance, the more scan points there are, and therefore, more scanning time. The scanning time can be calculated with the data we show in response 2.6. We could estimate a mean scanning time of 2 min (the maximum would be 2:42 with an HDR image in color).

3D laser scanners present data at resolutions or precisions expressed at 10 m. In the case of the RTC360, with a measurement rate of up to 2 million points/second, the data is as follows: For high resolution, (3 mm@10 m); for medium resolution, (6 mm@10 m); for low resolution, (12 mm@10 m). If we want to make sure of these resolutions, the logical thing to do is establish a scanning plan with the scan points at 10 m. However, such a decision would mean that the scan with TLS would take much longer. The UNAV campus is 113 Ha., 70% are green spaces, 10% is occupied by buildings and the other 20% consists of car parks, roads, etc. Three calculation scenarios were established:

In (I) we set out to estimate a scanning plan with a benchmark distance of 10 m. 100 scan positions/Ha. were estimated in the green spaces, making a total of 7910 scan/pos. 31 buildings associated with the campus and its activities had to be modelled and simulated. Only the exterior geometry of the facades had to be captured. In all, 775 scans were estimated for the facade perimeters (an average of 25 scans per building). The other areas, (parking, roads, etc.) were not as important for the environmental evaluation of the model, although they take up a lot of surface area, and so a total of 678 scan/pos was estimated. The total scan positions was 9363 that, with a scanning time of 2:42 min, made for a total of

52 days of field work without including the displacements of the scanner from one position to another.

In scenario (II) we estimated a scanning plan at 33 m in the green spaces, with an estimate of 16 scan positions per Ha. with a total of 1265 scan/pos. Scan positions were planned at 30 m in the building perimeters, but with a minimum of 3 scans per facade. 258 scan/pos were calculated for the characteristics of the buildings. In all, 339 scan/pos were estimated for the rest. The total number of scan positions was 1862, which with a scanning time of 2:42 min makes a total of 10.5 days of work without including the displacements of the scanner from one position to another. This would mean over 2 weeks' work.

In scenario (III) we estimated a scanning plan in the green spaces at distances under 50 m, with an estimate of 8 scan positions per Ha. with a total of 632 scan/pos. The previous plan of 30 m with 258 scan/pos was maintained for the building perimeters. In all, 169 scan/pos were estimated for the rest. The total number of scan positions was 1059, which with a scanning time of 2:42 min makes a total of 6 days field work. Displacements of the scanner from one scan/position to another (a minimum estimate of 30 s) adds one more day. This makes for a total of 7 days. The estimates calculated for working days of 8 h/day, although in early April there was more than 12 h of natural light a day, which gave a degree of margin for contingencies in the 7 days.

After presenting the estimation of the reduction of scan process in the field, next section shows how the cloud processing results obtained with TLS can be improved. Since the LiDAR clouds already have basic information on which to add the TLS clouds, the error and overlap of the resulting clouds is not the same as when scanning a single building using only TLS techniques. The main accuracy parameters to upgrade are: set error, overlap, strength of link and cloud-to-cloud error. An example of the direct result after the scanning and pre-processing phase of Zone 2 of the UPV/EHU campus in DSS is shown (Figure 14).

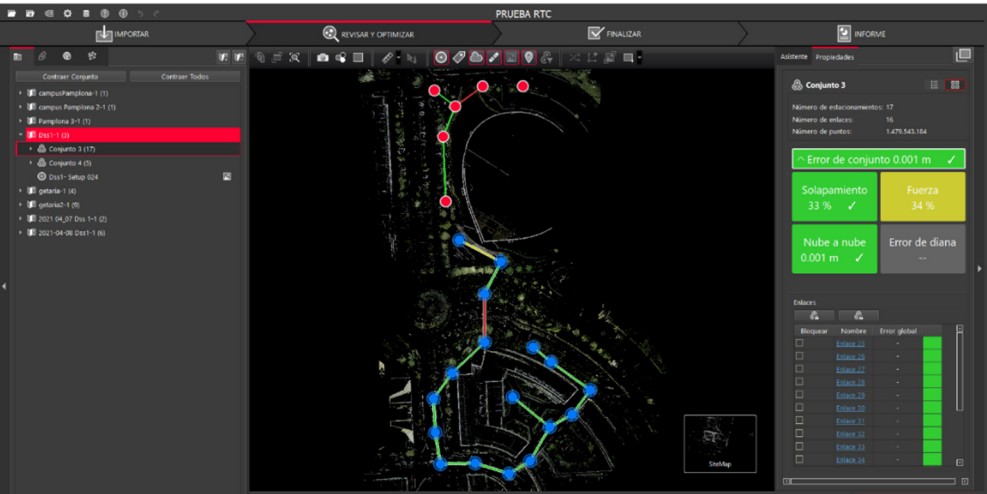

**Figure 14.** Review and optimization of the scan of the Donostia-San Sebastián campus in plan, using the Leica Cyclone Register, Zone 2.

As detailed previously, the processing phase has several stages separated in four tabs in the software. The first allows the data collected on-site to be imported. In this case, as pre-processing or pre-registration has been conducted, the scan points appear linked in the import from Cyclone FIELD 360 to Leica Cyclone REGISTER 360. At this point, the processing software analyzes the joint data in the field and assigns a color to each joint based on its strength and accuracy. Green indicates the highest strength and red the lowest strength; two other colors, yellow and blue, are in between. In the case of Figure 16, the input data for the processing marks the following results: set error 1 mm, overlap 33%, strength 34%, cloud-to-cloud error 1 mm. This data could be optimized in REGISTER 360 by optimizing the cloud-to-cloud joints, though the 1 mm assembly error is more than enough

to meet the needs of the 3D model in NEST. Furthermore, in environments with large vegetation, although the scan's accuracy is high (1–3 mm error), it may happen that the overlap between clouds is not as appropriate, due to the singularities of moving branches and leaves (Figure 15).

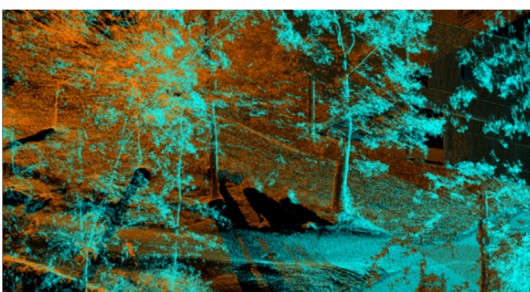

**Figure 15.** TLS LiDAR clouds in the tree-lined area of the UPV/EHU DSS campus. Forest biomass data can be obtained from these clouds, which offset the lack of ALS LiDAR clouds.

At the UNAV campus, the processing stage was very similar to that of the Donostia campus. Although, since the ALS LiDAR cloud is much more accurate, the scanning points per m$^2$ of campus are much lower. Considering also that 70% of the campus comprises green surfaces, means that there are no very reliable references between adjoining clouds. After the scan data dump before processing, the initial set therefore had much fewer joints in green (because of that lack of strength and overlap). Although the error was acceptable (2 mm), initially, the force was only 22% and the overlap 19%. Indeed, in some areas there is a previous joint that the software stopped linking so that it could be studied and improved during processing (Figure 16).

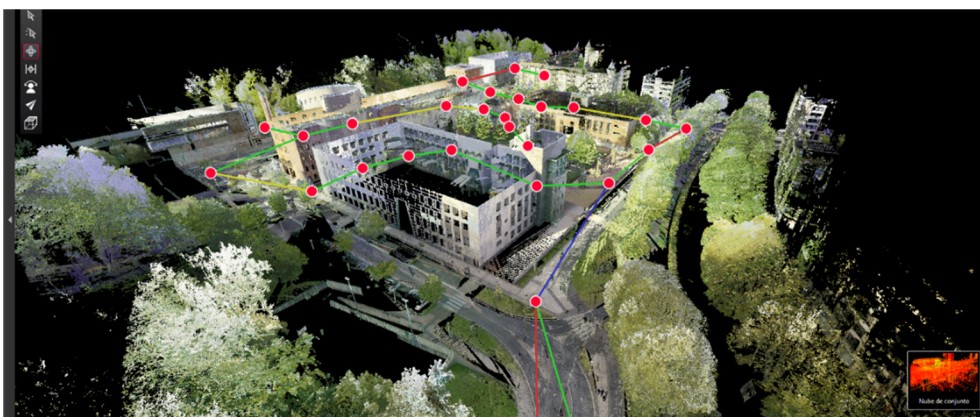

**Figure 16.** 3D point cloud. Review and optimization of scan of the UPV/EHU Campus, DSS, Zone 1. Description of the joints before the processing and optimization stage.

As the scan positions have been created at longer distances than usual to streamline the process, the pre-registration work conducted on site undergoes a revision. Since image 16 is in perspective, it is harder to see the color of the unions between scan points. When the data obtained in the field is imported, the software analyzes whether the points of a cloud of a scan position overlap enough with those of the previous and posterior clouds. The software checks and colors the unions in line with three concepts: Error, strength of union and overlap between clouds. If they have any undesirable parameters, the may color them in red, yellow or blue or it may not directly propose the union because it is outside a minimum range. What happens then, is that an optimization process commences analyses and improves the cloud-to-cloud union (as in Figure 8), between the two clouds that do not have a connection line in green (Figures 14 and 16). If we can improve the overlap parameters in this optimization process, the line of union changes to green and it is

accepted as valid. If we cannot change the union to green in the optimization process, there is still the option of carrying out another scan on site the next day and further strengthening the cloud of the set. This means that the field work should be processed every day, and so it is importance to use the tools mentioned in this article. The devices, software and applications enable pre-processing that greatly reduces the amount of daily processing work; they also make it easier to check that the on-site capture is satisfactory.

Once the processing stage is finished, to view the results and extract the necessary information the most appropriate file format is LGS. The free Leica Jetstream Viewer software enables the viewing and consultation of data from the digital model comprising the set of point clouds and the 360° images of each of the scan positions. This application allows a visual and metric inspection to be carried out virtually along the route. With this resource you can accomplish the analysis, verification and data extraction tasks needed for the modeling and simulation process in NEST, such as distances, areas, angles or even surface temperatures (Figure 17).

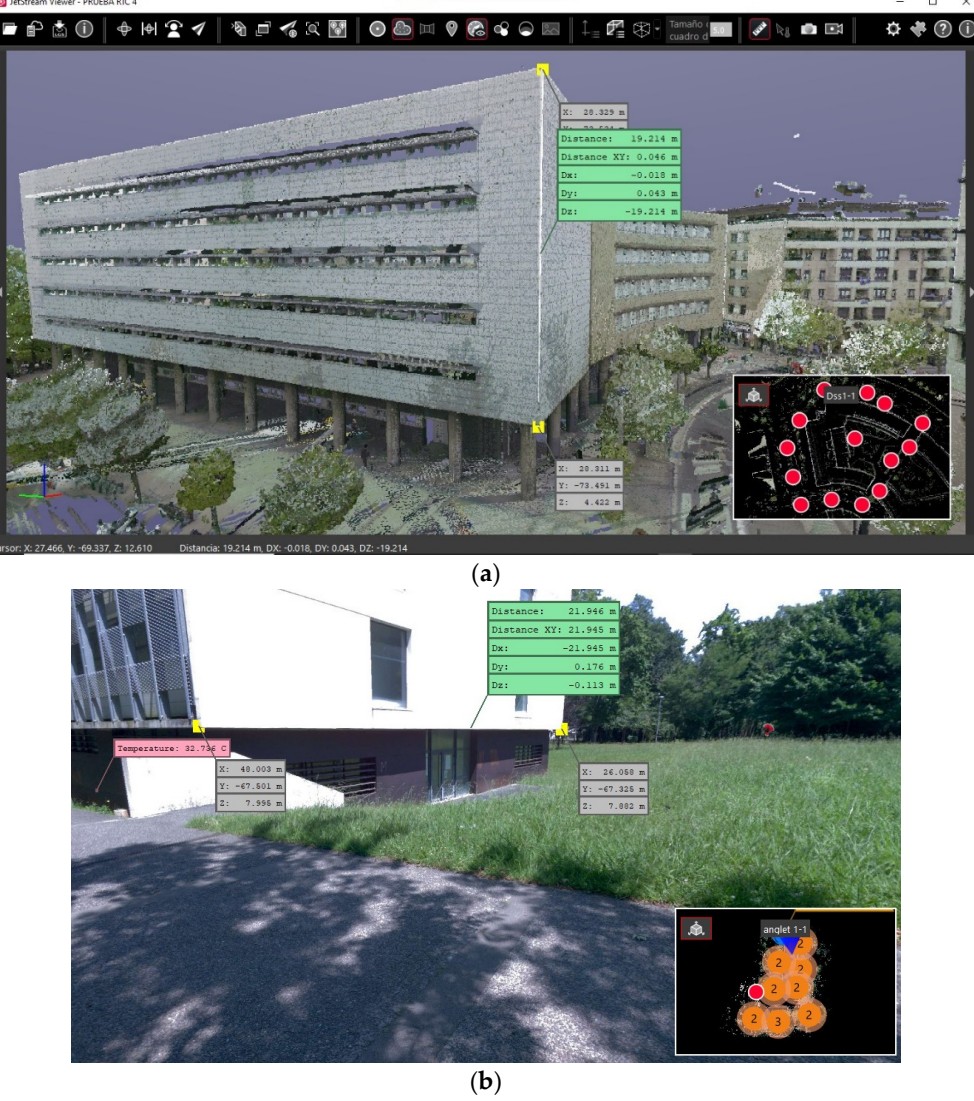

**Figure 17.** Visualization of results of the TLS LiDAR clouds in Jetstream Viewer, obtaining quantitative data for the model: (**a**) 3D colored cloud visualization, geometric data, Gipuzkoa School of Engineering building, UPV/EHU campus in DSS; (**b**) southwest façade of the same building. Visualization from the 360° image of the building, geometric data (green and gray) and thermal data (red).

Before starting to survey the campuses with TLS, the feasibility was analyzed in terms of resources and time needed to complete the ALS LiDAR cloud. TLS runtimes were calculated for comparison to UAV ones, with automated photogrammetry. In the case of the UPV/EHU campus in DSS, the configuration of the urban environment, its dimensions and the buildings' closeness enable very fast complementary capture with TLS, for which additional work of three days was estimated, in three zones. However, the UNAV campus in Pamplona presents groups of buildings, though with large distances between some of them. Capturing the entire campus with TLS would have taken several weeks, with an overwhelming amount of information. However, as the ALS LiDAR cloud, with a density of 14 pts/m$^2$, provided very acceptable measurement results, the complementary survey with TLS was estimated, after previous scanning plans, to take seven days, one day for each zone. After the TLS calculations, the resources were estimated with the UAV to decide which would be the fastest and most efficient combination.

### 4.3. Capture by UAV and SfM Photogrammetric Processing

The combination of data collection by UAV and the SfM automated photogrammetry method facilitates documentation due to the simplicity of the process and the width of the working range [78,79]. The study focuses on an assessment of the operation and efficiency of the capture for these large urban areas [14]. It concerns assessment of whether it could be faster and more efficient than additional captures with TLS to complete the ALS LiDAR cloud of the campuses. At the UPV/EHU campus, flight restrictions due to the location in controlled airspace made operations difficult. For that reason, it was ultimately decided to not complete the clouds with UAV-assisted photogrammetry.

The survey forecast was performed at the UNAV campus in Pamplona. The programmable application (DJI-GS Pro) allows configuration of the automatic flight mission through navigation based on satellite positioning (GNSS). The UAV used is a DJI Phantom 4 Pro model, equipped with a 1" CMOS sensor that reduces radial distortion and improves the metric quality of the SfM restitution method [80]. A grid comprising 147 square sectors measuring 100 m on each side and an area of 1 Ha was created, establishing a low-altitude flight parameter (37.3 m) to guarantee the model's accuracy [5]. This altitude offers a GSD factor of 1 cm/pixel with an overlap between photos, front and side, of 75%. The combined use of nadir shots (gimbal tilt—90°) and oblique shots (gimbal tilt—45°) was determined. One nadir and four oblique shots were planned per sector, the latter aligned with the four trajectories that join the vertices of each sector to its center (Figure 18).

The results of the resource feasibility study are presented in Table 6.

**Table 6.** Summary of UAV photogrammetry capture data.

| Sectors | Sectors Overlap | Flight Height | Ground Sample Distance GSD | Nadir Shot (−90°) | Oblique Shot (−65°) | Frontal Overlap | Lateral Overlap | Flight Time | Batteries | Photos |
|---|---|---|---|---|---|---|---|---|---|---|
| Sector type 1 ha | - | 37.3 m | 1 cm/pixel | 1 | 4 | 75% | 75% | 100 min | 5 sets | 450 |
| Set of sectors 147 ha | 10 m. (10%) | 37.3 m | 1 cm/pixel | 147 | 588 | 75% | 75% | 245 h | 735 sets | 66,150 |

Capturing data from the entire campus would require 245 flight hours, which, added to the preparation work, could mean a field task of more than one month. As previously justified, the additional works of the ALS LiDAR cloud with TLS techniques was estimated to take seven days, so the UAV was used to complement the TLS data in those sectors of the 147 planned sectors where there are no buildings.

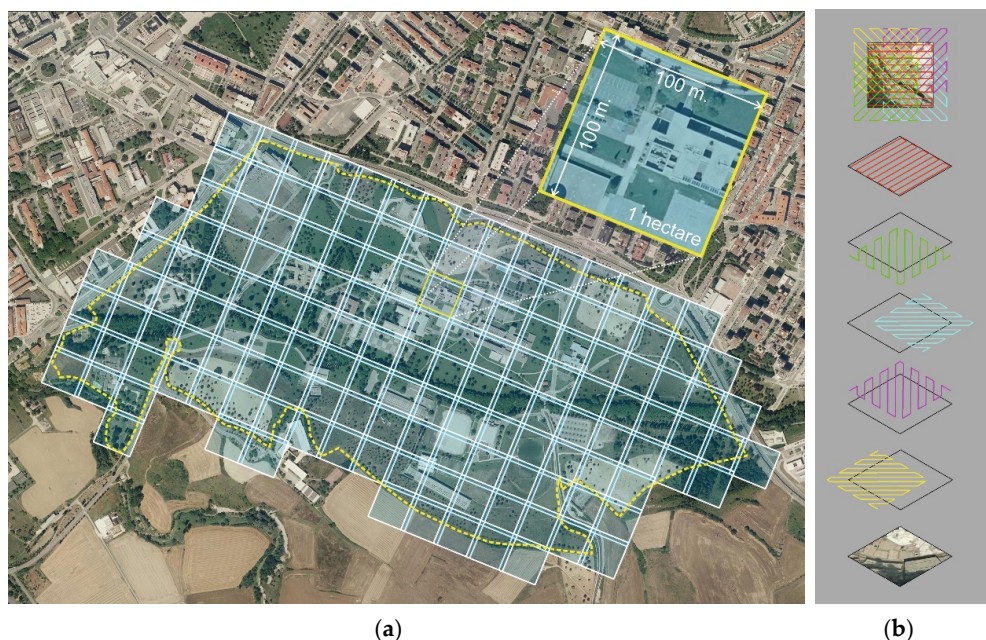

| (a) | (b) |

**Figure 18.** (**a**) Work planning prevision in 147 sectors; (**b**) planning of flight directions in one of the sectors, from top to bottom: 1. Shooting overlap, 2. Nadiral shooting/G.P.A.: −90°/C.A.: 0°, 3. Oblique shooting/G.P.A.: −45°/C.A.: 45°, 4. Oblique shooting/G.P.A.: −45°/C.A.: 135°, 5. Oblique shooting/G.P.A.: −45°/C.A.: 225°, 6. Oblique shooting/G.P.A.: −45°/C.A.: 315°, 7. Orthophoto, (G.P.A: Gimbal Pitch Angle; C.A: Course Angle).

### 4.4. Modeling

The point cloud of the ensemble will be used to perform the simulation 3D model of each campus in NEST (Figure 19).

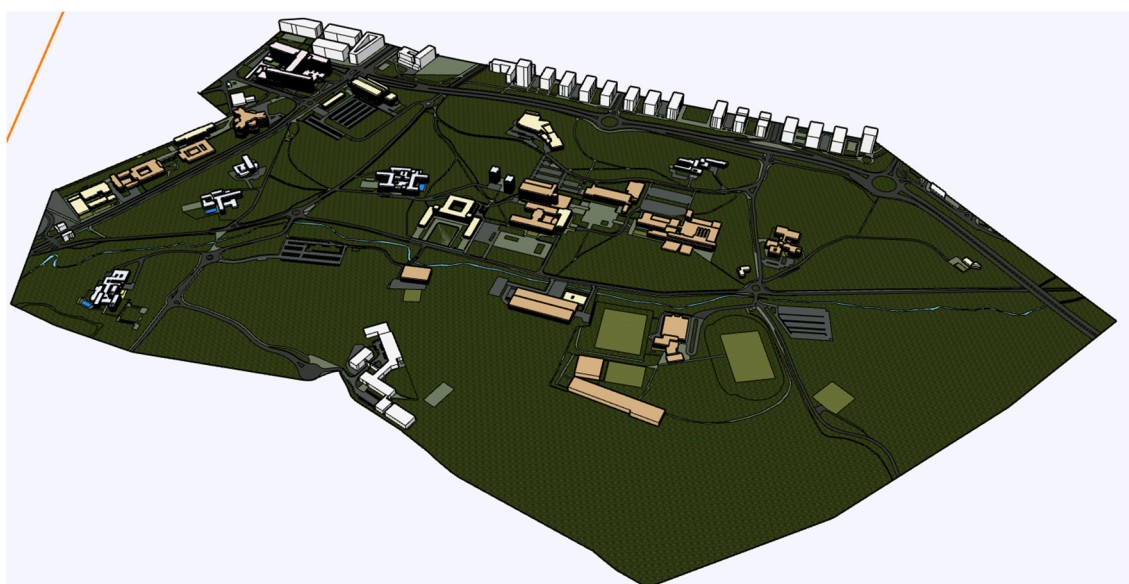

**Figure 19.** Final 3D model of the UNAV campus in Pamplona for simulation in NEST, (see Appendix A).

The resulting final cloud can also be segmented to model isolated buildings in greater detail and make specific simulations of the campuses (Figure 20).

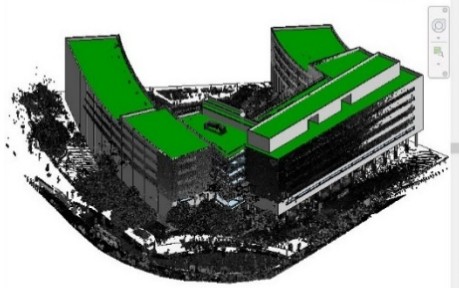

**Figure 20.** BIM model in Autodesk Revit software. School of Engineering building. Campus UPV/EHU DSS, Zone 2 of the scan.

### 4.5. NEST—Environmental Assessment Results

As previously stated, it is not the purpose of this article to analyze the results of the assessment, which has already been dealt with in other specific articles [11,12]. In any case, to round up the flow of the project, we felt it was appropriate to show a very reduced sample of the results given by NEST. An image of the simulation model and a summary table of the two campuses will be presented (Figure 21 and Table 7).

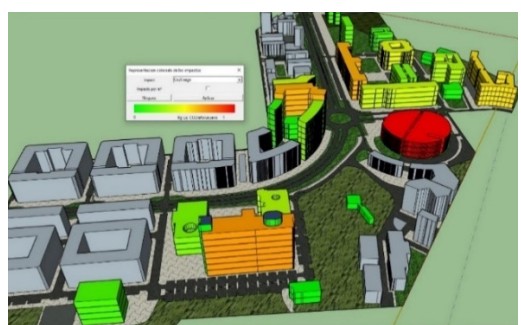

**Figure 21.** NEST model with simulation of $CO_2$ impacts. UPV/EHU DSS campus.

**Table 7.** Results obtained in the NEST simulation of the improvement scenarios for the years 2030 and 2050.

| Impact Indicator | Sector | Life-Cycle Stage * | Baseline Scenario | | 2030 | | 2050 | |
|---|---|---|---|---|---|---|---|---|
| | | | UNAV | UPV/EHU | UNAV | UPV/EHU | UNAV | UPV/EHU |
| PE (MJ/year) | Buildings (BP) | A1–3, A4–5, B2, B4, C1–4 | $7.5 \times 10^6$ | $1.0 \times 10^7$ | $1.7 \times 10^7$ | $1.5 \times 10^7$ | $3.4 \times 10^7$ | $1.7 \times 10^7$ |
| | Buildings (BU) | B6 | $1.3 \times 10^8$ | $3.8 \times 10^7$ | $1.1 \times 10^8$ | $3.7 \times 10^7$ | $9.3 \times 10^7$ | $3.5 \times 10^7$ |
| | Public lighting (PL) | B6 | $2.2 \times 10^6$ | $3.1 \times 10^6$ | $1.7 \times 10^6$ | $2.5 \times 10^6$ | $1.1 \times 10^6$ | $1.6 \times 10^6$ |
| | Mobility | A1–3, B6, C1–4 | $1.8 \times 10^3$ | $3.1 \times 10^3$ | $1.8 \times 10^3$ | $3.1 \times 10^3$ | $1.8 \times 10^3$ | $3.1 \times 10^3$ |
| GWP ($kg_{eq}CO_2$/year) | Buildings (BP) | A1–3, A4–5, B2, B4, C1–4 | $3.5 \times 10^5$ | $4.1 \times 10^5$ | $8.1 \times 10^5$ | $6.3 \times 10^5$ | $9.7 \times 10^5$ | $7.2 \times 10^5$ |
| | Buildings (BU) | B6 | $6.3 \times 10^6$ | $1.9 \times 10^6$ | $5.0 \times 10^6$ | $1.8 \times 10^6$ | $4.4 \times 10^6$ | $1.7 \times 10^6$ |
| | Public lighting (PL) | B6 | $2.3 \times 10^5$ | $3.3 \times 10^5$ | $1.9 \times 10^5$ | $2.7 \times 10^5$ | $1.2 \times 10^5$ | $1.7 \times 10^5$ |
| | Mobility | A1–3, B6, C1–4 | $1.0 \times 10^2$ | $1.4 \times 10^2$ | $1.0 \times 10^2$ | $1.4 \times 10^2$ | $1.0 \times 10^2$ | $1.4 \times 10^2$ |
| Energy consumption (kWh/year) | Natural gas (NG) | B6 | $1.7 \times 10^7$ | $1.5 \times 10^6$ | $9.4 \times 10^6$ | $1.2 \times 10^6$ | $5.7 \times 10^6$ | $6.9 \times 10^5$ |
| | Electricity (E) | B6 | $9.7 \times 10^6$ | $5.9 \times 10^6$ | $1.0 \times 10^7$ | $5.7 \times 10^6$ | $1.1 \times 10^7$ | $5.5 \times 10^6$ |
| | Biomass (B) | B6 | $7.0 \times 10^3$ | 0.0 | $6.3 \times 10^5$ | $4.5 \times 10^4$ | $1.0 \times 10^6$ | $9.8 \times 10^4$ |
| Renewable energy production (kWh) | Thermal solar (TS) | | $3.6 \times 10^4$ | $7.1 \times 10^4$ | $2.4 \times 10^5$ | $1.9 \times 10^5$ | $6.6 \times 10^5$ | $4.7 \times 10^5$ |
| | Photovoltaic (P) | | 0.0 | $4.7 \times 10^5$ | $1.8 \times 10^6$ | $7.8 \times 10^5$ | $3.7 \times 10^6$ | $1.6 \times 10^6$ |

* See Table 2.

## 5. Discussion

The starting hypothesis focused on the possibilities of obtaining rapid digitization of two university campuses that occupy a large area of the cities where they are located. The lockdown situation caused by COVID-19 did not allow long stays in the field to collect data, and a combination of techniques were studied so that the survey could be performed in just a few days.

After analyzing the results, it can be stated that it was possible to capture both university campuses in 3D, while field work time was reduced to just 10 days (7 + 3), with a satisfactory outcome that was economical and efficient. It was possible to complete the fieldwork in ten days, thanks to a combination of resources and techniques. After studying different technologies and resources, the work was accomplished by using ALS LiDAR point clouds captured by public services and by capturing complementary point clouds with greater density and accuracy by means of TLS; finally, UAV-assisted automated photogrammetric techniques were used to complement previous clouds. The study of each of these technologies used to achieve the objectives of this work enabled discussion of several specific advantages and disadvantages for the specific situation arising in this article.

The ALS LiDAR clouds obtained by public services are free and thus did not involve any investment of resources, finances or time. Although, it was discerned that those LiDAR clouds for the two cities in question have very different qualities, which affected the approach to the work and its development. The LiDAR cloud of the UNAV campus in Pamplona, with a density of 14 points/m$^2$, allows measurements to be made with sufficient precision to do the simulation model in NEST. However, the LiDAR cloud of the UPV/EHU campus in the city of Donostia-San Sebastián, with a maximum density of 2.2 points/m$^2$, has many limitations. Regarding the measurement of parts of building elements, it would be limited to dimensioning the total height of the built volume, without being able to make other kinds of measurements such as height of standard floor plan, opening sizes and even, in some cases, size of the façade surfaces. With respect to plant masses, it is also verified that the cloud's low density does not allow estimates of mass volume and in some cases, not even basic measurements of the plant element's height. That is why it was decided to complement the LiDAR data with massive point capture based on TLS, since that technique allows capture in the shortest possible time. Considering the characteristics of the LiDAR clouds found, the number of scanning points for each campus was more or less intensified. We have used this procedure to try out scan points at longer distances than usual (30 m, 50 m and 80 m, without exceeding the capture distance limit presented by the laser scanner, which is 130 m). It should be borne in mind that this work with TLS sets out to complement the LiDAR cloud of the Government of Navarra. With these tests we found that most scan points at distances of 30 m give acceptable results (unions in green) and that the maximum range of the scanner at high resolution is actually 60 m, even when the capacity is 130 m.

Regarding the complementary work of the point cloud with TLS techniques, the UPV/EHU campus has a higher scan position density than Pamplona, since its 2017 ALS LiDAR cloud is much less precise than that of the UNAV campus. The work with TLS showed some differences at each campus. Considering the maximum range of the scanner (130 m), a minimum of three scans per building façade could initially be considered. However, due to the characteristics of each campus, the pre-registration and processing work had different disadvantages. At the UPV/EHU campus, when joining the adjacent clouds, the pre-registration was strong enough because it was a flat area with high building density and little vegetation. Better cloud strength and overlap results were obtained, even though the scanner used the same millimetric precision. However, at UNAV, the campus's size makes the buildings more distant, so when scanning a building it is not easy to record many points from another adjacent building to strengthen the joints. In addition, the dense vegetation and the unevenness of the steep terrain make the overlap and strength of the final clouds lower. In any case, in Pamplona, this aspect was supplied with the good quality of the 2017 ALS LiDAR clouds.

The fact that there are large green areas without buildings on the UNAV campus imposes an added difficulty for quick capture with TLS. For that reason, before starting the fieldwork with TLS, it was evaluated whether UAV-assisted automated photogrammetry could be a faster and more efficient resource. Automated photogrammetry with UAV enables capturing the points of areas that cannot be accessed by the TLS scanner's laser beam, such as building roofs. Automation simplifies the process by reducing data collection times and offering a greater work range, besides enabling more efficient management of the capture process. However, the UAV, which is a cheaper device than the TLS, has several disadvantages for the challenge set out in this research. It requires certification with a professional pilot's license and its use is conditioned in multiple areas, both under airport control and over urban centers, where it is necessary to request authorizations well in advance. Moreover, in urban environments such as the UPV/EHU campus in Donostia-San Sebastián, ensuring flight safety over crowds of people and vehicles can be another handicap. In the case of the Pamplona campus, this last aspect does not affect it excessively, though its large size implied that data collection in the field would take more than four weeks, well above the forecasts for TLS. Use of the UAV was therefore limited to some sectors of the campus where there are no buildings.

## 6. Conclusions

Considering, on the one hand, the two case studies proposed (the university campuses of the UNAV in Pamplona and the UPV/EHU in Donostia-San Sebastián) and, on the other, the challenges that conditioned achieving the research objectives, three massive point capture resources were combined. The work started by using existing public ALS LiDAR clouds. Multiple technologies were then studied to reach the conclusion that the most appropriate technique for these two cases should focus on capturing LiDAR clouds with TLS. In specific cases, those two resources could be complemented by a massive capture of geometries and textures using UAV-assisted photogrammetry.

The ALS LiDAR point clouds captured by public services are an immediate and cheap resource, though their accuracy and efficiency depend on the quality of the devices used to capture them. Clouds with a density of $14\,\text{pts/m}^2$, captured with sensors such as those used by the Navarre government, allow measurements to be made on parts of building elements and vegetation. Below these densities, their usefulness for making such measurement has many limitations.

Regarding the complementary techniques proposed, some publications claim that UAV-assisted photogrammetry is a faster and more efficient technique than TLS. However, in this research, it has been possible to analyze and conclude that if the work starts by using ALS LiDAR clouds, then TLS may be a faster option to complete the final point cloud of the whole urban area. The results show that with points clouds overlapped with 360 images, produced with a combination of resources and techniques, it was possible to reduce the on-site working time by more than two thirds. TLS also has an advantage over photogrammetry since it allows 360° panoramic images overlapped with the point cloud. Taken together, this single file in .lgs format, with the two resources (points cloud + 360 image), creates an immersive experiences in the office that also makes it possible to know the campus in greater detail and extract all the data: points coordinates, distances, areas, angles and temperatures, directly in the 360° image with high quality resolution. The joint file facilitates detailed in-office study of the urban areas captured to make a 3D simulation model. In addition to fieldwork for the survey, visits to the field for other types of data collection in general can accordingly be reduced. In future research work, that digital model can be optimized in later stages until becoming a very effective digital twin (ref.). University campuses can be endowed with multiple sensors and measurers to monitor their environmental impacts, which can be recalculated in real time from the digital twin. This can enable optimization of the processes and work involved in managing university campuses [81–84].

The combination of ALS LiDAR clouds with TLS scanning clouds made it possible to obtain the 3D model for the environmental assessment of university campuses with NEST. The digitization of these two urban environments has been able to be conducted with a field job executed in a very short time, moving the work mainly to the office, with online collaboration and adjusting to the mobility restrictions imposed due to COVID-19.

**Author Contributions:** Conceptualization, I.L. and J.J.P.; methodology, I.L. and M.S.; software, J.J.P., I.L. and A.C.; validation, M.S., A.C. and J.J.P.; formal analysis, I.L. and A.C.; investigation, I.L. and J.J.P.; data curation, J.J.P., I.L. and M.S.; writing—original draft preparation, I.L. and J.J.P.; writing—review and editing, M.S., I.L. and A.C.; visualization, M.S. and A.C.; supervision, I.L. and J.J.P.; project administration, I.L.; funding acquisition, I.L. All authors have read and agreed to the published version of the manuscript.

**Funding:** This research was funded by the Nouvelle-Aquitaine/Euskadi/Navarre Euro-region (AECT). Project co-financed through the second session of the 2019 AECT call for projects.

**Institutional Review Board Statement:** Not applicable.

**Informed Consent Statement:** Not applicable.

**Data Availability Statement:** The data can be found on the collaboration platform of the University of the Basque Country (https://ehubox.ehu.eus/login accessed on 20 January 2022) and are available for restricted access.

**Acknowledgments:** We would like to thank the University of Navarra and the Arquitectura AH Asociados studio for their work acquiring data from UNAV for the baseline inventory. Additionally, to Alba Arias, Xabat Oregi and Cristina Marieta for the work carried out in the research project.

**Conflicts of Interest:** The authors declare no conflict of interest.

## Appendix A

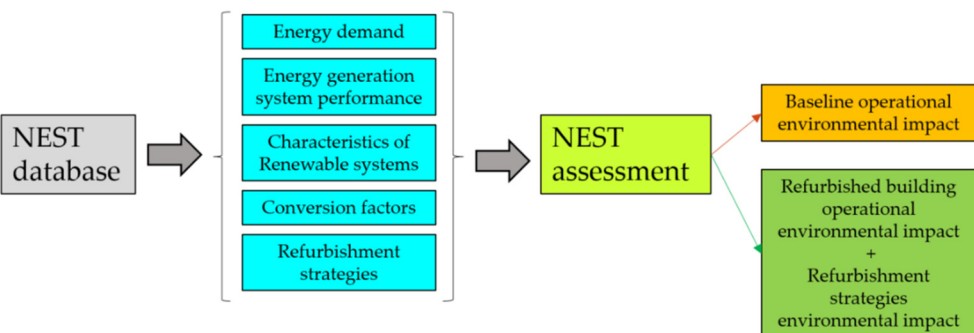

**Figure A1.** Diagram of the environmental assessment calculation processes of buildings in NEST.

Procedure of graphic tool NEST-Sketchup (Figure A2).

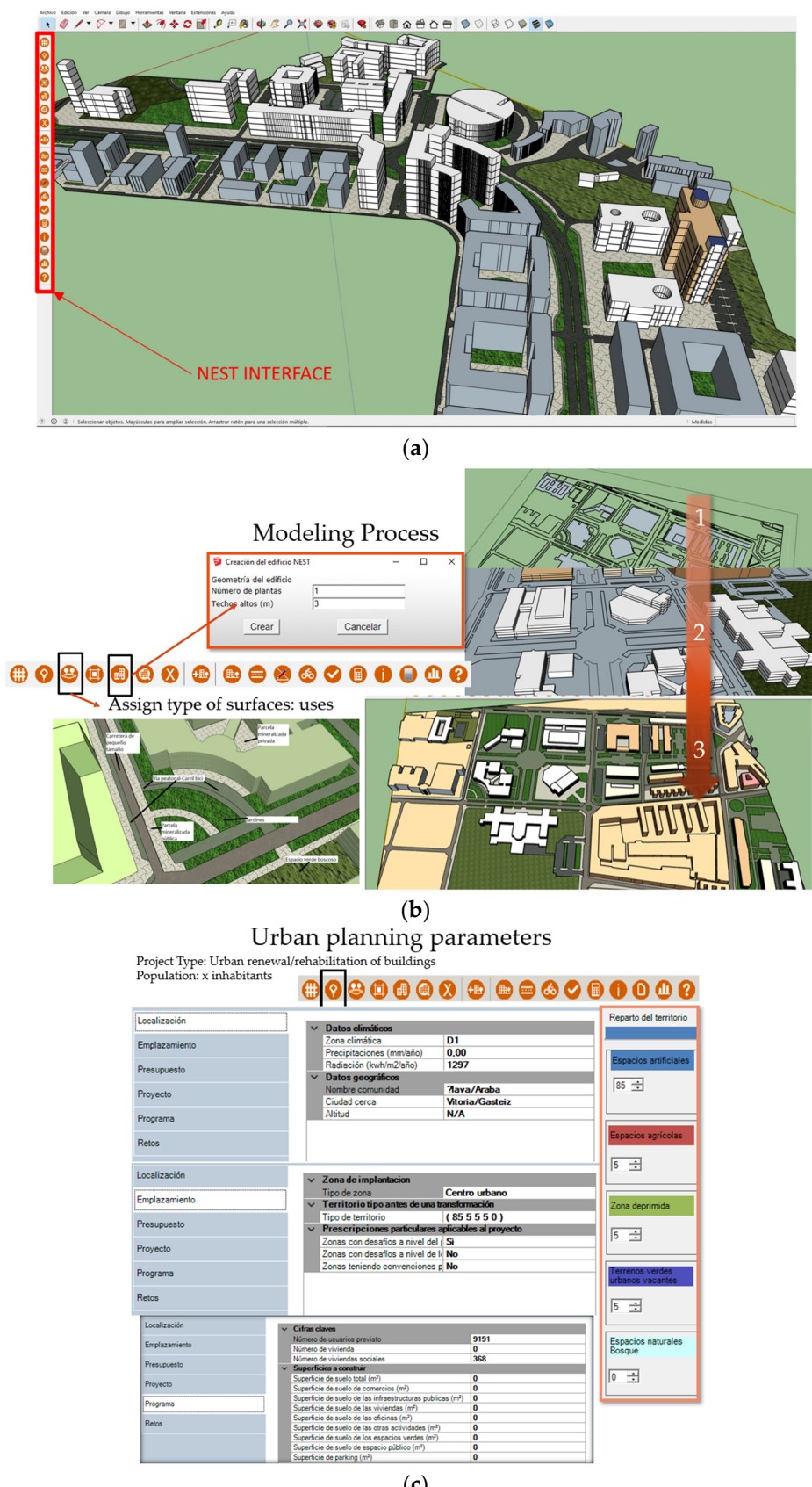

**Figure A2.** *Cont.*

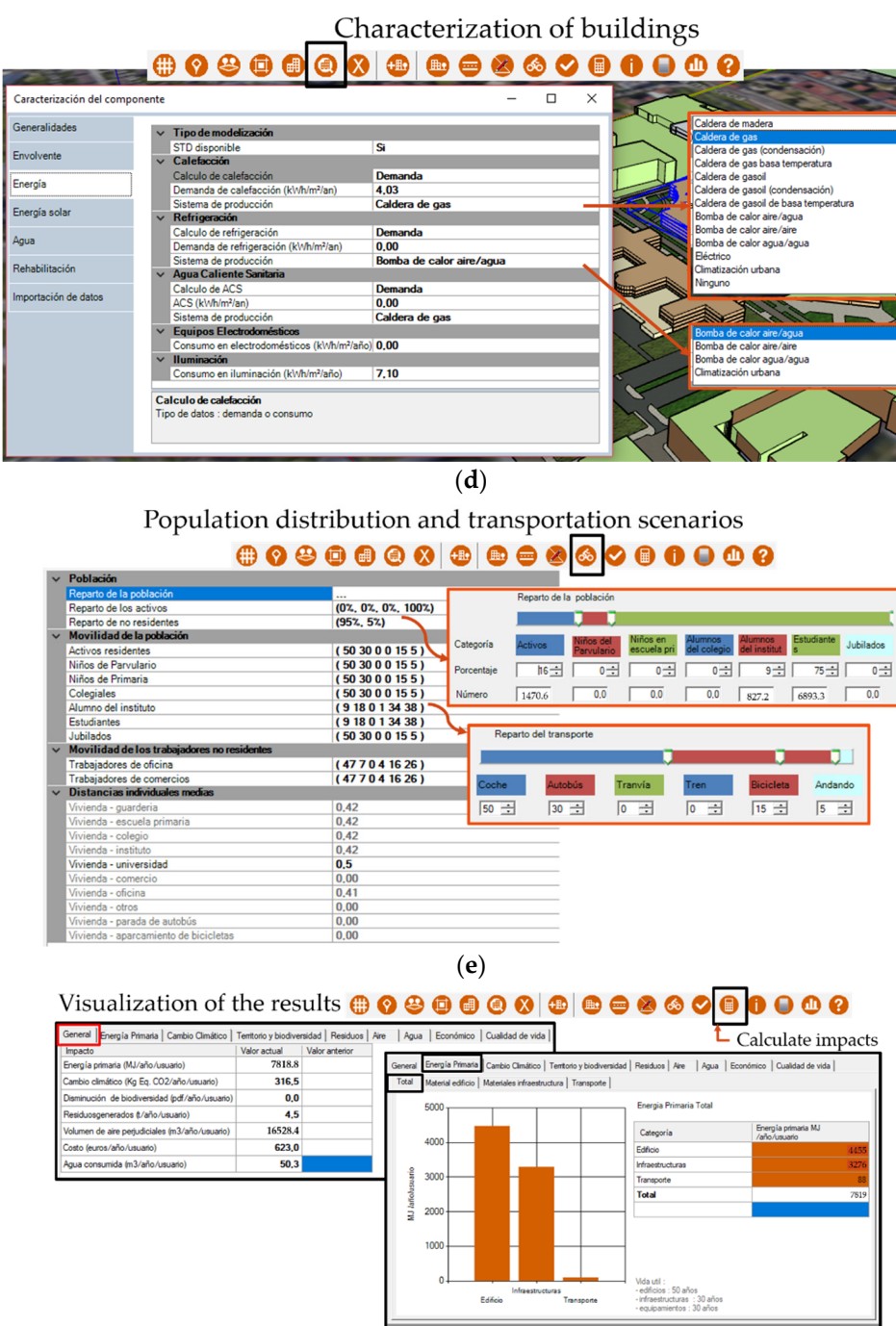

**Figure A2.** NEST-Sketchup procedure: (**a**) NEST Interface in Sketchup; (**b**) modeling process; (**c**) urban planning parameters; (**d**) characterization of buildings; (**e**) population distribution and transportation scenarios; (**f**) visualization of results [85].

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
