# Peer review of "Field Work’s Optimization for the Digital Capture of Large University Campuses, Combining Various Techniques of Massive Point Capture"

_buildings, doi:10.3390/buildings12030380_

Round 1

Reviewer 1 Report

Dear authors,

thank you for a newer version of the document. The article is quite improved now. The title is more appropriate and the abstract is better now. Here are my additional comments:

  1. English needs some polishing. There are again Spanish terms in the manuscript (i.e. line 224 Figures 6 y 7). Table 2: Infraestructure instead of infrastructure
  2. Figures 6 & 7 are almost unreadable. I understand that you were using ALS LIDAR in 2017. Did you consider using new scans in 2021? New technologies are available daily. I'm using L1 from DJI and with a short flight, I can have not just 2.2 points/m2, but 40 points/m2 very easily. It is obvious that with 2.2 points/m2 you cannot see anything.
  3. lines 371-375. I understand that you want to show as much data as possible, but sometimes some information about the technology is just too much. Try to shorten the paragraph or delete these sentences.
  4. Response 2.6 - I possess the same device as you were using: BLK360. I do not possess RTC. Line 377: With these scanners, in all cases, the HDR 360 image is captured in just 1 minute. For BLK360 when combining it with laser scanning this is not true. Also, in the same paragraph, you are mentioning the 6mm@10m precision for RTC. This is the same precision as you can get with BLK360. My concern is that you are not showing precision on larger distances. In an example, for BLK360 it states that the precision is 6mm @ 10m but  8mm @ 20m. As you were capturing the data from larger distances this information is crucial. Also, you are stating: With these scanners, in all cases, the HDR 360 image is captured in just 1 minute. Yes, HDR image but not the full scan.
  5. Point 2.7. Your words: Any tablet or mobile phone that uses iOS or Android can be used for this purpose. I disagree. I'm asking you which tablet did you use? Please write it.
  6. line 522: please cite the PhD thesis
  7. line 540 and all others referring to the ISO norms. It is not 14,040 ISO Standardization but 14040 ISO. Also, cite it.
  8. I'm not familiar with the ISO norm. Please check if the same norm should be in the text and in the table. If not, then explain the norm EN15978 in the text.
  9. line 682: I like your optimistic enthusiasm: a minimum estimate of 30 seconds. I agree it can be very fast but this is really fast. Also, you did not calculate if you have the scan place you need to retake and for sure this must happen from time to time.

In the end, "I still like the content". The topic is interesting and needs investigation. My only concern is that you are giving too much space to technical data of hardware and software - try to reduce these parts. Also, try to answer on other bullet points that I wrote.

The last remark, you wrote: We actually studied the LiDAR points clouds obtained by the L1 and, after comparing them with the free LiDAR clouds of the Government of Navarra, captured with the Leica Single Photon LiDAR (SPL100), we did not find any notable differences. The mentioned statement is not in the text but I must answer that it cannot be true. We are using L1 extensively lately and I must say that probably we will stop using BLK360 because of it. The precision is great and it can not be compared with SPL100. I understand that it is expensive, I'm not saying that you must buy it. In my opinion, for the expertise you have done it can be a time-saver.

Reviewer 2 Report

The following minor changes needs to addressed for the possible publications

  1. The manuscript requires a moderate English corrections. Perform the grammatical check for the manuscript,
  2. The work described in the present study can be impactful. But in the present era of Industry 4.0, the purpose of present study can be satisfied by using the emerging technologies such as Digital Twin, Image processing, Machine learning, etc. Kindly refer the articles as mentioned below, but not limited to these

A. Xingzhi Wang, Yuchen Wang, Fei Tao, Ang Liu, New Paradigm of Data-Driven Smart Customisation through Digital Twin, Journal of Manufacturing Systems, Volume 58, Part B, 2021, 270-280, 0278-6125, https://doi.org/10.1016/j.jmsy.2020.07.023.

B. Lee, A.; Lee, K.-W.; Kim, K.-H.; Shin, S.-W. A Geospatial Platform to Manage Large-Scale Individual Mobility for an Urban Digital Twin Platform. Remote Sens. 2022, 14, 723. https://doi.org/10.3390/rs14030723

C. Pal S.K., Mishra D., Pal A., Dutta S., Chakravarty D., Pal S. (2022) Image Processing for Digital Twin. In: Digital Twin – Fundamental Concepts to Applications in Advanced Manufacturing. Springer Series in Advanced Manufacturing. Springer, Cham. https://doi.org/10.1007/978-3-030-81815-9_4

D. Warke, V.; Kumar, S.; Bongale, A.; Kotecha, K. Sustainable Development of Smart Manufacturing Driven by the Digital Twin Framework: A Statistical Analysis. Sustainability 2021, 13, 10139. https://doi.org/10.3390/su131810139

E. Kumar, Satish; Patil, shruti; Bongale, Arunkumar; Kotecha, Ketan; Bongale, Anup kumar M.; and Kamat, Pooja, "Demystifying Artificial Intelligence based Digital Twins in Manufacturing- A Bibliometric Analysis of Trends and Techniques" (2020). Library Philosophy and Practice (e-journal). 4541.

3. Describe the future scope for the proposed work with the help of above mentioned articles, so as to implement the digital twin for the field work optimization.

Round 2

Reviewer 1 Report

Thank you for the additional work. The article looks fine now. The review is anonymous so I cannot add some of my papers :).

This manuscript is a resubmission of an earlier submission. The following is a list of the peer review reports and author responses from that submission.

Round 1

Reviewer 1 Report

Dear authors,

thank you for presenting the idea of digital capture of urban environments. The manuscript deals with the on-site capturing of two university campuses and compares the results obtained by different means of capturing data. 3D capturing of the built environment is very popular, and new technologies er on the market on a weekly basis. The same applies to the software dealing with producing digital twins. The topic of the paper is in the scope of the Journal Buildings

Although, the paper deals with an interesting concept, it still lacks some scientific soundness. The manuscript is written more like a technical report than a scientific paper. Here are my remarks.

  1. The Title: Efficient Digital Capture of Urban Environments for Environ-2 mental Evaluation and Improvement by Means of 3D Models 3 with Information - Please reformulate it. It sounds very pretentious and with a big breakthrough in the field of environmental evaluation but it is not. The paper deals with case studies of two Universities nad environmental evaluation is a very small part of the paper.
  2. Abstract: This study focuses on...3D model for two university campuses. The study cannot be focused on 3D modeling. This is not a scientific paper. This European project... - please delete it from the abstract. Somebody will read this in 5 years and this is not crucial information for the abstract.
  3. English: there are several "strange" interpretations in the document. In I.e. please do not start sentences with "This". Please improve it a little bit.

        line 440: Software y

       lines 460-461- the text is in Spanish

       line 661: this is possible. What is possible?

  1. Introduction part: First two paragraphs should be at the end of the Introduction part, not at the beginning. Please describe more in detail what is NEST. Also, put a reference to the original document for NEST.
  2. line 235: ...at each parking lot. You are mentioning parking lots a lot, why? Either I don't understand the English expression, either parking lot is completely irrelevant.
  3. lines 261-262: I don't understand the whole paragraph
  4. line 281: I personally possess the Leica devices and recently we are using them a lot. Also, we are complementing them with DJI UAV. Also, I was believe it or not several times at the UNAV campus in Pamplona. I cannot believe that laser scanners have problems with uneven terrain. You just need to have better overlaps.
  5. line 276: Because the situation generated by the COVID-19 health crisis meant that movement 276 was very limited... From my point of view, this is not a problem but a benefit. There are no people there who are "generating" noise and the capturing should be easier.
  6. line 307: ...in a short time... If you need to have the really good quality it is not so short time. HDR images and high-density point clouds take almost 8 minutes per position. Also, with bigger distances, the quality is of course not so significant. To have distances more than 30 m from one filling position to the other one is not so good.
  7. line 316: which tablet?
  8. line 403: flight restrictions are not of great concern. If you are an "official" drone operator then with a permit you can fly wherever you want
  9. line 427: NEST-Sketchup. Please show and explain the procedure
  10. Figure 18: review and optimization? I don't see the optimization
  11. Figure 20: b) where is b)?
  12. line 629: 7 days is very optimistic. Please show how did you calculate it.
  13. Figure 21: please show the procedure and explanation of how did you get.
  14. Chapter 4.5 should be the core of your paper but it is not even one page. Table 6 is not understandable without explanations. What are A1-3, B6, etc.
  15. line 690: I don't believe that 3 scans per building facade are enough.
  16. line 738: since it allows 360... With photogrammetry, you can easily do it.
  17. References: please reference all of the mentioned software

In the end, the paper is written like a technical report, not as a scientific paper. I like the content but don't see a huge scientific contribution. Also, recently, DJI placed L1 LIdar on market. Everything you've done here can be done with it in a shorter period of time. Laser scanner, UAV photogrammetry is combined and all of the gathered data can be easily processed.